# Contextualized Visual Personalization in Vision-Language Models

Yeongtak Oh [* 1]  Sangwon Yu [* 2]  Junsung Park [1]  Han Cheol Moon [3]  Jisoo Mok [4]  Sungroh Yoon [1 5]

## Abstract

Despite recent progress in vision-language models (VLMs), existing approaches often fail to generate personalized responses based on the user's specific experiences, as they lack the ability to associate visual inputs with a user's accumulated visual-textual context. We newly formalize this challenge as contextualized visual personalization, which requires the visual recognition and textual retrieval of personalized experiences by VLMs when interpreting new images. To address this issue, we propose CoViP, a unified framework that treats personalized image captioning as a core task for contextualized visual personalization and improves this capability through reinforcement-learning-based post-training and caption-augmented generation. We further introduce diagnostic evaluations that explicitly rule out textual shortcut solutions and verify whether VLMs truly leverage visual context. Extensive experiments demonstrate that existing open-source and proprietary VLMs exhibit substantial limitations, while CoViP not only improves personalized image captioning but also yields holistic gains across downstream tasks. These results position CoViP as a key step toward robust and generalizable contextualized visual personalization. Project page: https://oyt9306.github.io/covip.github.io/

## 1. Introduction

Recent advances in vision-language models (VLMs) (Liu et al., 2023; Li et al., 2024; Chen et al., 2024; Bai et al., 2025b) have demonstrated impressive performance across a wide range of vision-language tasks, including image captioning, visual question answering, and open-ended visual dialogue. Despite these advances, however, current VLMs remain limited in their ability to personalize visual understanding based on user-specific context (Wang et al., 2025b; Bai et al., 2025a). For example, while a VLM may correctly recognize a person in an image as a man wearing a black suit, it typically fails to identify that the person corresponds to the user's brother mentioned in prior interactions. This issue indicates that existing VLMs do not yet possess a genuine capability for *visual personalization*.

To address this limitation, a growing body of work has explored methods for enhancing visual personalization in VLMs (Alaluf et al., 2024; Nguyen et al., 2024; Kim et al., 2025a). While existing methods successfully enable personalization over simple attributes or identities, they remain limited in scope. In particular, they do not account for personalization grounded in rich, experience-level user context, such as past interactions or episodic memories, which remains largely underexplored in existing literature (Hao et al., 2025; Nguyen et al., 2025; Oh et al., 2025; Hong et al., 2025; Doveh et al., 2025).

In this work, we define such visual experiences as a *contextual history* that integrates previously observed images with associated personal textual information. We consider a setting in which VLMs maintain a user's past interactions in their context and are expected to leverage this history when interpreting new visual input. We refer to this realistic setting as **contextualized visual personalization**. Figure 1 illustrates the example of the use-case for contextual visual personalization in VLMs.

In practice, personalization in real-world settings is inherently diverse and open-ended, as it depends on implicit user intent and fine-grained contextual cues. Consequently, relying solely on task-specific post-training is both insufficient and inefficient for achieving robust personalization, since it does not scale to the long tail of personalized, context-dependent behaviors in real-world interactions. To address this challenge, we propose a unified approach to contextualized visual personalization that targets a shared underlying process common across downstream tasks. We argue that this process naturally aligns with the objective of personalized image captioning, which focuses on grounding visual

---

[*]Equal contribution [1]Department of Electrical and Computer Engineering, Seoul National University, Seoul, South Korea [2]Upstage AI, South Korea [3]Samsung Electronics, South Korea [4]DGIST, South Korea [5]Interdisciplinary Program in Artificial Intelligence, Seoul National University, Seoul, Korea. Correspondence to: Jisoo Mok <jmok@dgist.ac.kr>, Sungroh Yoon <sryoon@snu.ac.kr>.

*Proceedings of the 43^{rd} International Conference on Machine Learning*, Seoul, South Korea. PMLR 306, 2026. Copyright 2026 by the author(s).

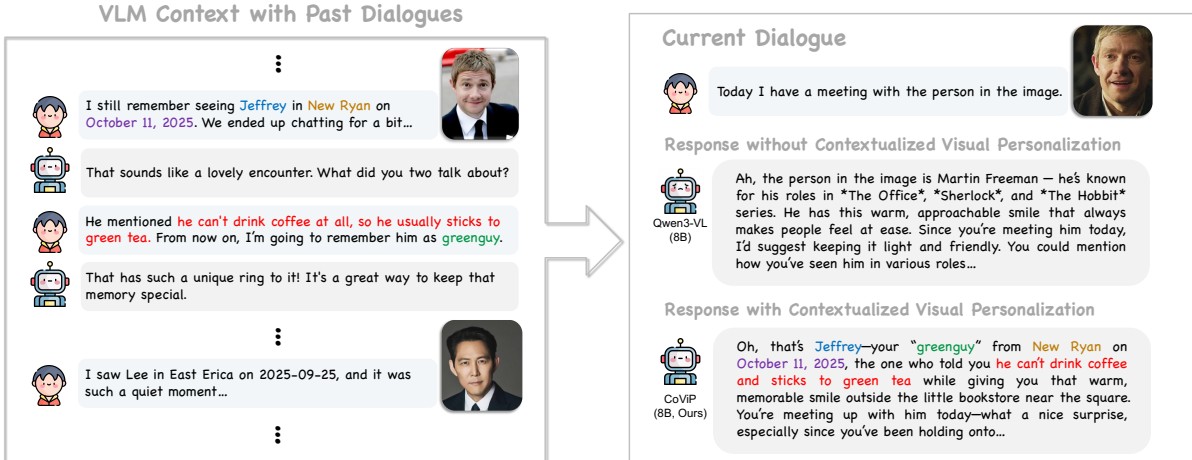

*Figure 1.* Qualitative example of the use-case for contextual visual personalization in VLMs. Note that our CoViP effectively responds to the question while integrating the mentioned personal details from the given multimodal contexts.

inputs in user-specific contextual knowledge, without requiring additional task-specific processing. Accordingly, we leverage personalized image captioning as a proxy task to effectively model and learn this shared process.

Building on this insight about personalization, we introduce **CoViP** (**Co**ntextualized **Vi**sual **P**ersonalization via Image Captioning), a unified framework that enables holistic personalization by explicitly modeling this underlying process. CoViP formulates contextualized visual personalization as a personalized image captioning task, in which a VLM recognizes relevant visual concepts in a query image, retrieves the corresponding user-specific context, and directly incorporates the retrieved information in the generated caption. Within this framework, we first construct a novel and challenging personalized image captioning benchmark that faithfully captures the complexities of contextualized visual personalization. We then adopt a reinforcement learning (RL)-based post-training strategy to optimize VLMs for producing personalized captions through our captioning benchmark. At inference time, we further introduce *caption-augmented generation (CAG)*, where the VLM's own generated caption is reused as an explicit conditioning signal to guide personalized response generation.

Beyond modeling and training, we place particular emphasis on evaluation, as contextualized visual personalization should avoid spurious textual shortcuts (*e.g.*, directly retrieving query-related hints from context without recognition) that allow VLMs to answer questions while bypassing visual understanding. To this end, we design a suite of *diagnostic downstream tasks* that explicitly assess whether a VLM correctly recognizes and retrieves personalized visual context. These diagnostics span personalization from reactive to proactive settings, enabling fine-grained analysis of personalization capabilities in realistic settings.

Our experiments demonstrate that CoViP effectively im-

proves personalized image captioning performance and leads to consistent gains in contextualized visual personalization. Additionally, we observe that proprietary VLMs exhibit unstable behavior on our diagnostic tasks, underscoring the need for an explicit post-training stage for captioning prior to downstream adaptation to achieve robust and generalizable personalization. Moreover, the VLM trained with CoViP yields superior results across all proposed diagnostic tasks, indicating holistic gains in personalization capability. Finally, we show that CAG further amplifies performance during inference by capitalizing on the fine-grained details of the generated caption.

Our contributions are summarized as follows:

- We introduce a novel paradigm of contextualized visual personalization, formalizing personalization as the ability to leverage user-specific visual experience in contextualized visual understanding.

- We present CoViP, a unified framework that operationalizes this concept through personalized image captioning, RL-based post-training, and caption-augmented generation.

- We design diagnostic evaluation tasks that systematically assess contextualized visual personalization from reactive to proactive settings, enabling a systematic analysis of limitations of existing baselines.

- Experimental results demonstrate that CoViP consistently improves personalization performance across our benchmark and various diagnostic tasks.

## 2. Previous Works on VLM Personalization

Early approaches to personalized VLMs (Alaluf et al., 2024; Nguyen et al., 2024; 2025; An et al., 2025) primarily leverage the inherent zero-shot capabilities of off-the-shelf models to handle explicitly defined concepts retrieved from exter-

*Table 1.* Comparison of personalization paradigms. We contrast task-limited personalization with contextualized visual personalization under interleaved image–text contexts and a dedicated benchmark.

| Method | Post-Training | Multi-Concept | External VLM | Interactive-Dialogues | Long-Contexts | Generalize | Use case | Evaluation |
|---|---|---|---|---|---|---|---|---|
| MyVLM | ✗ | ✗ | ✗ | ✗ | ✗ | ✗ | Cap/ VQA | Name recall |
| Yo'LLaVA | ✗ | ✗ | ✗ | ✗ | ✗ | ✗ | Cap/ VQA | Name recall |
| TAME | ✗ | ✗ | ✓ | ✓(1-turn) | △ | ✗ | VQA | VQA accuracy |
| RAP | ✓(SFT) | ✓ | ✗ | ✗ | △ | ✗ | Cap/ VQA | Name recall |
| RePIC | ✓(RL) | ✓ | ✗ | ✗ | △ | ✓(Cap) | Cap | Name recall |
| **CoViP (Ours)** | **✓(RL)** | **✓** | **✗** | **✓(3-turns)** | **✓** | **✓(Tasks $\geq$ 3)** | **Cap/ VQA** | **CapEval-QAs** |

nal databases. Recently, post-training-based methods have emerged to enable VLM personalization in an in-context manner; for instance, Hao et al. (2025) integrates retrieval with personalized generation, while Oh et al. (2025) further advances this direction by achieving robust performance in multi-concept image captioning. Another emerging line of research incorporates contextual information to facilitate personalization. Doveh et al. (2025) enhances VLMs to recognize the same objects in a query image and its context, and Kim et al. (2025a) introduces an evaluation benchmark when the same single object appears in both the context and the query. Hong et al. (2025) proposed a training-free method for long-context personalized conversations, while Mei et al. (2026) formulated a personal memory assistant pipeline for retrieving and reasoning over user-specific experiences across heterogeneous sources.

In Table 1, we further clarify how our approach differs from existing baselines. We exclude direct comparisons with MyVLM (Alaluf et al., 2024) and Yo'LLaVA (Nguyen et al., 2024), as these methods do not support long in-context personalization settings. Moreover, since TAME (Hong et al., 2025) relies on additional memory controlled by an external VLM, we also exclude it from direct comparison and focus our evaluation on post-training-based baselines.

Overall, prior efforts largely focus on *explicit personalization*, evaluating models based on their ability to retrieve surface-level attributes (*e.g.*, names) from context. Such settings overlook deeper semantic reasoning and the nuanced nature of human–model interaction. In contrast, we study a more realistic and challenging setting of *implicit personalization* in interleaved multimodal contexts, where personalization cues must be inferred from visual experience, as shown in Figure 1. To this end, we propose a post-training pipeline that enhances contextual reasoning and personalized generation beyond surface-level retrieval.

## 3. CoViP: Contextualized Visual Personalization via Image Captioning

### 3.1. Problem Definition

We introduce the notion of *contextualized visual personalization*, which refers to a VLM's ability to generate person-

alized responses by jointly reasoning over a visual input and a user-specific interaction history. Formally, given a query image $x$, a user prompt $p$, and a context $c$ that contains the past visual-textual interactions, a VLM $f$ with parameters $\theta$ is expected to generate a response

$$y = f_\theta(c, x, p). \tag{1}$$

Here, $c$ represents user experiences accumulated through prior user–model interactions. Accordingly, the generated response $y$ should reflect these personalized experiences, rather than producing a generic description.

A key challenge in this setting arises from the open-ended nature of the user prompt $p$. Since $p$ can vary arbitrarily across tasks and interaction scenarios, the output space of $y$ becomes extremely large. As a result, directly optimizing task-specific objectives for downstream outputs is insufficient for achieving robust personalization, as it is infeasible to exhaustively control all possible personalization behaviors through supervised training alone.

To this end, we posit that diverse personalization tasks share a common underlying process. Specifically, regardless of the downstream task, a VLM must first *interpret the query image in the context of the user's past experiences* before producing a response. To explicitly model this process, we decompose the internal mechanism into two stages:

$$z = h_\theta(c, x), \qquad y = g_\theta(z, p), \tag{2}$$

where $h_\theta$ serves as a contextual visual encoder that grounds $x$ and $c$ into a personalized latent representation $z$, and $g_\theta$ acts as a task-specific generator that generates the final response conditioned on both $z$ and $p$, all within a single VLM. Crucially, while $g_\theta$ varies significantly across tasks as determined by $p$, the personalization-critical component $h_\theta$ is shared. This observation motivates us to focus on learning $h_\theta$ to holistically improve contextualized visual personalization.

Furthermore, we observe that $h_\theta$ is inherently aligned with the objective of *personalized image captioning*. As captioning is a fundamental generation task that avoids extraneous reasoning (*e.g.*, thinking) steps, the resulting caption $s$ directly reflects the model's user-specific contextual under-

standing. Accordingly, we adopt captioning as a reliable proxy for externalizing the latent personalization state $z$.

Based on this insight, we propose leveraging personalized image captioning as a proxy task to learn contextualized visual personalization. By optimizing models to generate captions that faithfully reflect a visual context, we aim to indirectly, yet effectively, improve personalization performance across various downstream tasks. The following sections describe how we construct a personalized image captioning benchmark, how we post-train VLMs through this benchmark, and how, at inference time, the generated captions are reused to guide personalized response generation.

### 3.2. Personalized Image Captioning Benchmark

We propose a challenging benchmark that captures the complexity and realism of *contextualized visual personalization*, as depicted in Figure 2. Our benchmark evaluates a VLM's ability to implicitly infer personalization cues from interleaved multimodal contexts, while requiring accurate perception of each visual concept. Overall, the benchmark constitutes 2.8K training samples and 1.3K test samples.

#### 3.2.1. DATASET CONSTRUCTION

**Image generation and quality filtering.** To construct the benchmark, we use an image-generative VLM (Comanici et al., 2025) to synthesize a controlled set of images. We first curate a foundational image database from research-permissible, open-source repositories (*e.g.*, Unsplash[1]), and then design diverse interaction scenarios among concepts involving humans, objects, and animals. Based on these scenarios, we generate query images with varying visual complexity by including one to four concepts per image. Representative multi-concept query images are shown in Figure S.3.

To ensure data reliability, we apply additional quality filtering using a text-generative VLM (Comanici et al., 2025). This model evaluates instruction adherence by verifying whether each query image is generated in accordance with the prompt, and assesses visual faithfulness by checking whether the positive concept images used to generate a query image actually appear in the resulting image. Samples exhibiting prompt inconsistencies or visual artifacts are removed through this filtering process. The detailed descriptions of the quality filtering criteria and procedures are provided in Appendix D and Table S.12.

**Dialogue generation.** To ensure the verifiability of the constructed contextual memory, we generate multi-turn dialogues that simulate free-form interactions between a user and the model, while strictly grounding all utterances in factual information related to a specific image. Here, fac-

tual information includes concrete locations, timestamps, events, or scenarios, while deliberately excluding subjective preferences, personal habits, or hallucinated content. This design choice enables reliable, unambiguous evaluation of the generated captions, as the correctness of each description can be directly verified against the grounded facts in the dialogue context. The prompt templates used for dialogue generation are provided in Table S.14.

**Construction of image–text interleaved contexts.** To construct interleaved image–text contexts, we incorporate both positive and negative samples within each context. Here, each sample consists of an image and its associated dialogue. A positive image corresponds to a ground-truth concept image used to synthesize a query image, while a negative image is visually similar to a positive one but does not appear in the query image.

To construct visually fine-grained yet discriminative images within the context, we utilize a retrieval process to explicitly control visual granularity by selecting negative images that are visually similar to the positive ones. Specifically, using the CLIP-L/14 vision encoder (Radford et al., 2021), we retrieve the top-2 most similar images for each positive image based on cosine similarity. These retrieved images are then paired with generated dialogues to form negative samples, which are collectively used with positive samples to compose the interleaved context. An illustration of a constructed multimodal context is provided in Figure S.1.

#### 3.2.2. EVALUATION PROTOCOL VIA CAPTION-BASED MCQA PROBING

To evaluate the degree of personalization in generated captions, we introduce a caption-based multiple-choice question answering (MCQA) probing protocol termed *CapEval-QAs* that assesses whether a caption correctly reflects contextually relevant information while excluding irrelevant content. Specifically, given a dialogue context $c = [d_1, \ldots, d_N]$, we construct factual MCQA pairs $(q_{ik}, a_{ik}) \sim \mathcal{G}(d_i)$ using an LLM-based generator $\mathcal{G}$, where each dialogue yields three QA pairs grounded in its factual content, that is, $k = 1, 2, 3$.

During evaluation, a judge model $\mathcal{J}$ is provided only with the generated caption $s$ and is asked to answer each question $q_{ik}$. For $(q_{nk}, a_{nk})$ pairs derived from the dialogue of positive concepts $d_n$, $\mathcal{J}$ is expected to answer correctly, and the corresponding accuracy is reported as *Positive Accuracy*, denoted by $\mathbf{Acc}^+$, which measures how precisely the caption captures relevant contextual information. Conversely, for $(q_{mk}, a_{mk})$ pairs derived from the dialogue of negative concepts $d_m$, $\mathcal{J}$ is expected to respond with uncertainty, reflecting the absence of such information in the caption. Performance in this setting is reported as *Negative Accuracy*, denoted by $\mathbf{Acc}^-$, which quantifies the model's ability to avoid incorporating irrelevant contextual details.

---

[1]https://unsplash.com

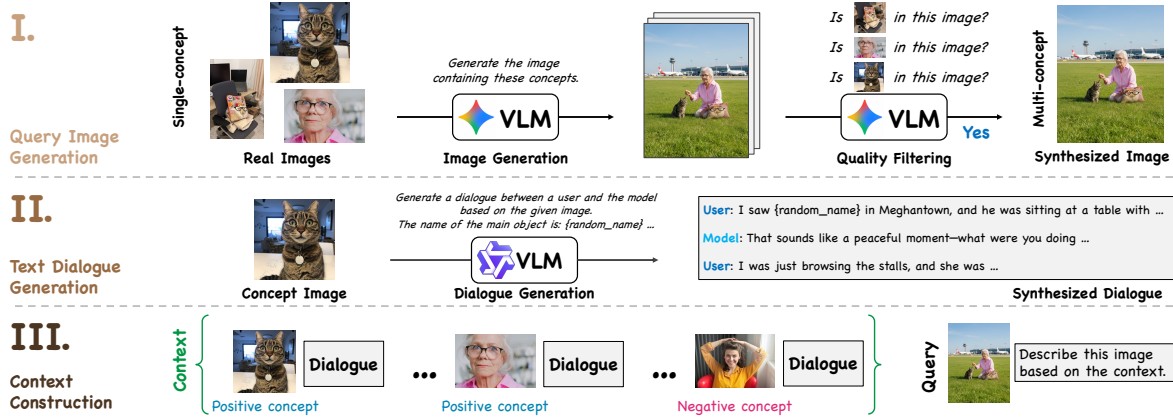

*Figure 2.* Illustration of the proposed personalized image captioning benchmark construction.

The prompt templates for generating and evaluating these MCQA pairs are provided in Tables S.15 and S.17. The validity of this protocol is further confirmed by human evaluation results in Appendix F. Pseudocode and a detailed version of the evaluation protocol description are presented in Appendix B.

## 3.3. Post-Training for Personalized Image Captioning

Building upon the benchmark introduced in Section 3.2, we propose an RL-based post-training framework for personalized image captioning.

### 3.3.1. OBJECTIVE FORMULATION

Following the objectives outlined in Section 3.1, our task is to generate a faithful caption $s$ conditioned on a query image $x$, context $c$, and captioning prompt $p_s$. Let $\pi_\theta(s \mid x, c, p_s)$ denote the captioning policy induced by VLM. We post-train the model parameters $\theta$ by maximizing the expected verifiable reward (VR). Formally, we optimize

$$\max_\theta \quad \mathbb{E}_{(x,c)\sim\mathcal{D}_{\text{tr}}} \mathbb{E}_{s\sim\pi_\theta(\cdot|x,c,p_s)}\Big[r(s,x,c)\Big] \quad (3)$$

where $r(s,x,c)$ is a VR designed to explicitly reinforce (i) *recognition* through fine-grained visual discrimination among in-context visual concepts and (ii) *retrieval* by encouraging context-groundedness in the generated captions. Concretely, we decompose the reward as follows:

$$r(s,x,c) = \underbrace{r_{\text{vis}}(x,c)}_{\text{recognition}} + \underbrace{r_{\text{caps}}(s,c)}_{\text{retrieval}}. \quad (4)$$

We post-train the policy $\pi_\theta$ to maximize the expected VR $r(s,x,c)$ of Eq. (4) during optimizing Eq. (3) using GSPO (Zheng et al., 2025) algorithm. Detailed descriptions for post-training are presented in Appendix C.

### 3.3.2. VERIFIABLE REWARD DESIGN

**Set-level recognition VR.** To augment fine-grained visual perception in VLMs, we introduce a novel F1-based VR tailored to concept recognition within a context. In each context, several concept images are provided along with a query image, and the model is required to identify which concepts appear in the query. The model outputs only the indices of the matched concepts, yielding a predicted index set $\hat{H}$ (*e.g.*, $\lceil 1, 3, 5 \rceil$). We then compute a dense set-level F1 score between $\hat{H}$ and the ground-truth set $H$:

$$r_{\text{vis}}(x,c) = \text{F1}(\hat{H}, H) = \frac{2\,|\hat{H} \cap H|}{|\hat{H}| + |H|}, \quad (5)$$

where $\text{TP} = |\hat{H} \cap H|$, $\text{FP} = |\hat{H} \setminus H|$, and $\text{FN} = |H \setminus \hat{H}|$. This F1-based VR provides meticulous feedback: the model receives a partial reward when only a subset of relevant concepts is identified. Predicting irrelevant indices increases FP and reduces precision, while missing relevant indices increases FN and reduces recall. An illustration of a multi-image example within a context is presented in Figure S.11.

**MCQA-based retrieval VR.** We incentivize the model to generate a caption $s$ that encapsulates sufficient dialogue-derived evidence required to resolve the MCQA tasks in Section 3.2.2. Let $\mathcal{J}(\psi(s,q))$ be the judge's selected choice for $s$ and question $q$ following the evaluation prompt template $\psi(\cdot)$, where the QA pairs $(q, a)$ are pre-generated from $c$. We construct a set of positive questions $\{q_k^+\}_{k=1}^K$ for positive images with corresponding correct answers $a_k$, as well as a set of negative questions $\{q_\ell^-\}_{\ell=1}^M$ for negative images whose correct choice is $D$ (*i.e.*, *"The answer cannot be determined"*). We can formulate the captioning VR by leveraging $(s, c)$ to $(s, QA)$ indirectly as

$$r_{\text{caps}}(s,c) = \begin{cases} -1, & \text{R}(s) > 0, \\ \sigma^+(s; QA^+) - \sigma^-(s; QA^-). & \text{otherwise.} \end{cases} \quad (6)$$

Here, $R(\cdot)$ serves as a degeneration filtering indicator, while $\sigma$ quantifies the accuracy score evaluated by an external LLM: $\sigma^+(s, QA^+) = \sum_k \mathbb{I}\big[\mathcal{J}(\psi(s, q_k^+)) = a_k^+\big]$ and $\sigma^-(s, QA^-) = \alpha \sum_\ell \mathbb{I}\big[\mathcal{J}(\psi(s, q_l^-)) \neq D\big]$. In these expressions, $\alpha$ is a scalar coefficient and $\mathbb{I}[\cdot]$ is the indicator function. This contrastive construction rewards responses that correctly answer positive questions while penalizing responses that induce non-$D$ answers on negative questions. The detailed description of the degeneration filtering indicator is provided in Appendix C.

### 3.4. Caption-Augmented Generation

We introduce a novel inference-time strategy to enhance the performance of contextualized visual personalization on downstream tasks. We term this approach *Caption-Augmented Generation (CAG)*, in which the model first synthesizes a descriptive caption and subsequently leverages it as a conditioning signal for downstream personalization. Specifically, instead of directly generating the output response $y$, the model first generates a caption $s$ and then conditions on it to produce the final output as follows:

$$s \sim \pi_\theta(\cdot \mid x, c, p_s), \quad y \sim \pi_\theta(\cdot \mid x, c, p_d, s), \quad (7)$$

where $p_s$ and $p_d$ denote the prompts for the captioning and downstream tasks, respectively.

## 4. Diagnostic Evaluation of General Personalization Capability

To evaluate contextualized visual personalization beyond proxy training objectives, it is necessary to assess whether a model can correctly leverage user-specific multimodal context when responding to realistic downstream queries. However, existing benchmarks for visual personalization (Nguyen et al., 2024; Kim et al., 2025a; Oh et al., 2025) do not capture whether a model reflects fine-grained, episodic user information embedded in contextual memories. To address this gap, we introduce three diagnostic tasks designed to evaluate contextualized visual personalization under realistic interaction scenarios. Each task probes the model under challenging conditions that explicitly preclude shortcut behaviors, such as answering solely based on dialogue context without grounding the visual input in user-specific experiences. Figure 3 shows a visualization of examples in diagnostic tasks. Design details on the downstream tasks are provided in Appendix D.

**Last-Seen Detection (LSD).** LSD evaluates whether a model can recognize the individual in a query image and *identify the most recent encounter with that person* from the user's contextual history. Given that the context contains multiple interactions involving the same individual, the model must retrieve all relevant entries and perform temporal reasoning to determine the correct answer. This task,

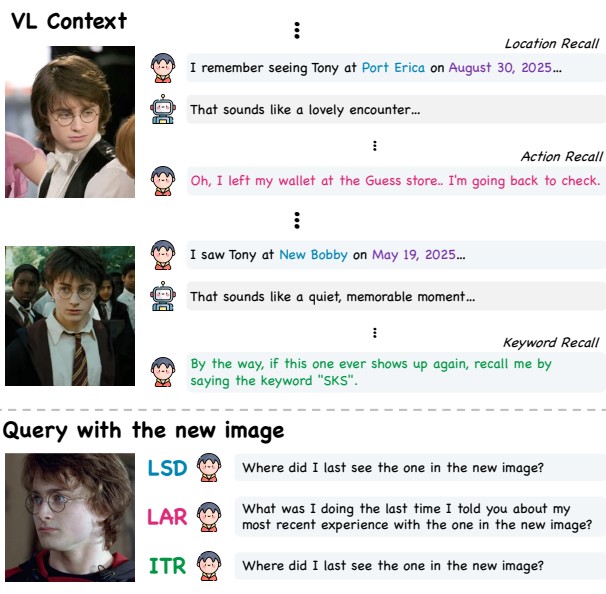

*Figure 3.* Visualization of diagnostic personalization tasks.

therefore, requires grounding visual input in user-specific history rather than relying on partial matches or surface-level textual cues.

**Last-Action Recall (LAR).** LAR extends LSD by requiring the model not only to identify the most recent encounter with the person in the query image, but also *to retrieve the fine-grained action described in that interaction*. Specifically, the model must first perform temporal reasoning to locate the dialogue corresponding to the latest encounter, and then extract what the user was doing at that time. As a result, LAR evaluates whether the model can go beyond explicitly specified temporal reasoning and actively retrieve fine-grained episodic information in response to loosely formulated user instructions, such as implicitly asking what the user was doing at the time.

**Instruction-Triggered Recall (ITR).** ITR evaluates proactive personalization, where the model is expected to surface relevant personalized information even when it is not explicitly requested (Zhang & Sundar, 2019). In this task, the context includes a past instruction indicating that *the user wishes to be notified with a specific keyword (e.g., "SKS") upon encountering a particular individual again*. When presented with an image of that individual, the model must proactively recall this instruction and incorporate the keyword into its response, even though the user does not explicitly request it in the current turn. This evaluates the model's ability to trigger personalized behavior based on implicit contextual cues rather than explicit queries.

## 5. Experiments

**Baselines.** We evaluate performance on our benchmark against proprietary VLMs, including non-thinking mod-

*Table 2.* CapEval-QAs performances (described in Section 3.2.2) on our personalized image captioning benchmark. Here, $\Delta$ denotes the performance gain relative to the base VLM, and the VLM post-trained with CoViP shows superior performance.

| Models | 1-Concept | | 2-Concepts | | 3-Concepts | | 4-Concepts | |
|---|---|---|---|---|---|---|---|---|
| | $Acc^+$ | $Acc^-$ | $Acc^+$ | $Acc^-$ | $Acc^+$ | $Acc^-$ | $Acc^+$ | $Acc^-$ |
| Proprietary VLMs (Close-sourced) | | | | | | | | |
| GPT-4o | 34.2 | 98.2 | 21.6 | 98.6 | 20.4 | 99.3 | 15.3 | 99.2 |
| GPT-5 | 48.3 | 97.3 | 28.2 | 97.9 | 26.1 | 98.7 | 18.9 | 98.7 |
| Gemini-2.0-Flash | 41.9 | 96.7 | 28.6 | 97.3 | 26.6 | 98.3 | 23.1 | 98.3 |
| Gemini-3.0 Pro | 58.1 | 96.6 | 45.1 | 97.2 | 39.0 | 98.3 | 32.4 | 97.9 |
| Open-Sourced VLMs | | | | | | | | |
| Qwen3-VL-8B | 39.0 | 97.5 | 25.6 | 97.7 | 23.3 | 98.1 | 18.6 | 98.1 |
| Qwen3-VL-30B-A3B | 40.2 | 96.2 | 27.5 | 97.7 | 25.3 | 97.7 | 20.1 | 98.1 |
| Post-Training-based Personalized VLMs | | | | | | | | |
| Qwen3-VL-8B + RAP | 20.5 | 99.0 | 10.4 | 99.1 | 9.9 | 99.5 | 7.3 | 99.2 |
| Qwen3-VL-8B + RePIC | 44.0 | 97.1 | 31.7 | 97.0 | 29.2 | 97.8 | 24.0 | 97.2 |
| Qwen3-VL-8B + CoViP | **77.4** | 94.8 | **68.4** | 94.1 | **65.2** | 94.8 | **59.7** | 92.8 |
| $\Delta$ (Increased) | + 38.4 | - | + 42.8 | - | + 41.9 | - | + 41.1 | - |

*Table 3.* Recall score performances on the downstream diagnostic personalization tasks.

| Models | LSD | | LAR | | ITR | |
|---|---|---|---|---|---|---|
| | Direct | w/ CAG | Direct | w/ CAG | Direct | w/ CAG |
| Proprietary VLMs (Close-sourced) | | | | | | |
| GPT-4o | 28.7 | 33.6 | 4.80 | 7.40 | 8.40 | 13.5 |
| GPT-5 | 28.5 | 34.4 | **50.8** | **59.3** | 18.6 | 10.5 |
| Gemini-2.0-Flash | 52.7 | 46.0 | 11.6 | 42.3 | 66.1 | 12.2 |
| Gemini-3.0 Pro | **76.2** | **89.3** | 9.40 | 44.0 | **89.4** | 19.0 |
| Open-Sourced VLMs | | | | | | |
| Qwen3-VL-8B | 29.8 | 48.8 | 17.4 | 19.6 | 9.40 | 6.80 |
| Qwen3-VL-30B-A3B | 25.6 | 42.1 | 7.60 | 16.8 | 8.80 | 0.40 |
| Post-Training-based Personalized VLMs | | | | | | |
| Qwen3-VL-8B + RAP | 27.0 | 28.8 | 1.40 | 0.80 | 0.00 | 0.20 |
| Qwen3-VL-8B + RePIC | 32.7 | 52.1 | 16.2 | 17.8 | 27.2 | 27.8 |
| Qwen3-VL-8B + CoViP (Ours) | 37.2 | 58.2 | 34.8 | 49.2 | 28.0 | **42.8** |
| $\Delta$ (Increased) | + 7.4 | + 9.4 | + 17.4 | + 29.6 | + 18.6 | + 36.0 |

els such as `ChatGPT-4o` and `Gemini-2.0-Flash`, and thinking models such as `ChatGPT-5` (OpenAI, 2025) and `Gemini-3.0-Pro` (Google, 2025). In addition, among post-training-based personalization methods, we compare against RAP (Hao et al., 2025) and RePIC (Oh et al., 2025).

**Post-training setup.** For a fair comparison, we retrain all baselines using the same `Qwen3-VL` backbone and report the reproduced results. Training is conducted using LoRA (Hu et al., 2022), and details are provided in Appendix C.

### 5.1. Benchmarking Personalized Image Captioning

In Table 2, we report results for each concept in the test set of our benchmark. The CapEval-QAs evaluation protocol assesses whether VLMs successfully perform personalized image captioning, with MCQA accuracy serving as the pri-

mary metric. Here, positive accuracy measures the correctness of MCQ answers for positive concepts using only the generated caption. Negative accuracy, in contrast, measures the model's ability to correctly determine that questions associated with negative concepts cannot be answered based solely on the generated caption.

> **Key Finding 1**
>
> Existing VLMs lack the ability to generate context-grounded captions.

Interestingly, we observe that both open-source and proprietary VLMs exhibit a limited ability to generate context-grounded captions. Among proprietary models, `Gemini-3.0-Pro` achieves the strongest performance. Furthermore, existing post-training-based personalization base-

lines provide only marginal gains on our benchmark: RAP even underperforms the zero-shot baseline, due to limited generalizability, as also noted by Oh et al. (2025), while RePIC yields only a slight improvement. These findings suggest that prior approaches fail to adequately incorporate useful contextual information when generating personalized captions in our benchmark.

> **Key Finding 2**
>
> CoViP substantially improves the VLM's contextual grounding capability through RL-based post-training.

We analyze the performance gains achieved by RL-based post-training relative to the base VLM. Notably, our approach yields an average performance improvement of approximately 40% across all concepts. Appendix F provides additional discussion on the $\mathrm{Acc}^-$ results.

## 5.2. Evaluations on Downstream Personalization Tasks

We evaluate the generalization capability of CoViP on downstream personalized tasks and compare it against competitive baselines. As shown in Table 3, although Gemini-3.0-Pro achieves strong performance on both the ITR and LSD tasks, it underperforms CoViP on the LAR task. In contrast, the GPT-5 attains the best results on the LAR task but exhibits substantially weaker performance on the LSD and ITR tasks. RePIC shows only marginal or no improvement over the base VLM across all tasks. By comparison, CoViP consistently delivers stable and notable improvements over the base VLM across all diagnostic tasks, with the gains becoming even more pronounced when CAG is applied. We draw the following core observations.

> **Key Finding 3**
>
> Personalized image captioning provides a reliable bridge for downstream personalization by enabling CoViP to effectively leverage CAG.

**1) CoViP improves personalization of VLM across all downstream tasks.** Compared to baselines, CoViP exhibits consistent performance gains across all tasks. Specifically, CoViP substantially outperforms the base VLM, indicating that post-training tailored for personalized image captioning guides the model to generate context-grounded captions and strengthens its fundamental personalization capability.

**2) CoViP effectively benefits from CAG**. We observe that applying CAG to CoViP yields consistent, task-agnostic performance gains over direct zero-shot inference. Interestingly, while CAG improves performance on the LSD and LAR tasks for proprietary VLMs, it causes a significant performance drop on the ITR task, with recall decreasing to below 20% across all VLMs after applying CAG. This trend is also observed in open-source VLMs. We attribute this behavior

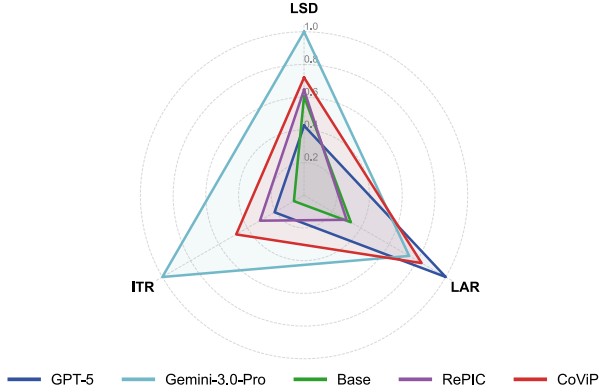

*Figure 4.* Visualization of comparative results for visual-triggered personalization diagnostics. Scores indicate relative performance.

to the challenging nature of the ITR task, which requires reliably capturing easily overlooked triggering keywords. Generic captions that lack fine-grained details fail to benefit from CAG and instead degrade performance. In contrast, CoViP incorporates granular contextual information from dialogues into its captions, enabling it to effectively leverage the benefits of CAG and achieve consistent gains.

**3) Prior baselines fail to generalize in diagnostic evaluations.** Post-training-based personalization baselines, such as RAP and RePIC, that show limited improvements on our benchmark also demonstrate limited generalization on diagnostic tasks compared to CoViP. This suggests that only CoViP equips VLMs with the capability to generalize to more complex personalization scenarios.

## 6. Discussions

**Why personalized image captioning should precede downstream personalization tasks.** As illustrated in Table 3 and Figure 4, proprietary VLMs exhibit relatively strong performance on diagnostic tasks, despite achieving substantially lower scores on the personalized image captioning benchmark compared to VLMs trained under the CoViP framework. However, this advantage of proprietary VLMs does not generalize across all diagnostic tasks, and performance gains obtained through CAG are often unstable and highly task-dependent (*i.e.*, ITR task). We attribute this behavior to a fundamental gap: proprietary VLMs do not reliably perform personalized image captioning. This limits the effectiveness of CAG, resulting in inconsistent and unreliable performance improvements in downstream applications. Therefore, to mitigate such instability and achieve robust improvements across diagnostic tasks, it is essential to incorporate a post-training stage focused on personalized image captioning prior to downstream inference.

**CoViP enhances context retrieval and integration rather than recognition itself.** Figure 6 analyzes the relationship

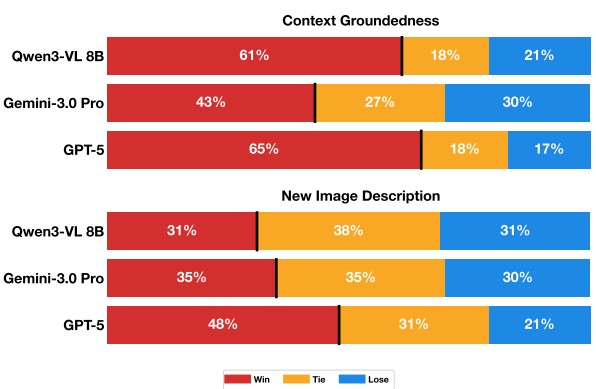

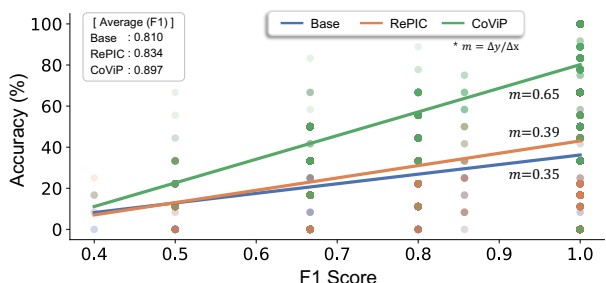

*Figure 5.* Results of the human preference evaluation. Here, Win denotes the win rates of CoViP compared to the baseline.

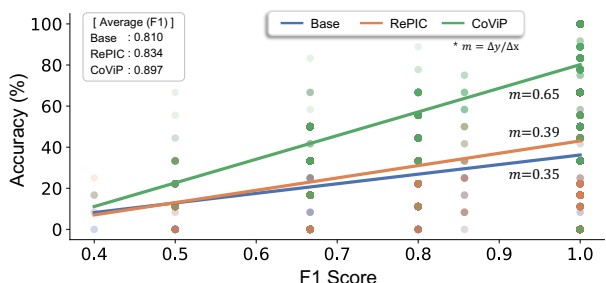

*Figure 6.* Scatter plot of recognition versus retrieval on the proposed benchmark. Recognition is measured by the F1 score of entity name inclusion between generated captions and ground-truth dialogues, while retrieval is measured by positive MCQA accuracy. Here, $m$ denotes the slope of the linear regression line.

between recognition and retrieval. As shown in the figure, the average F1 score exhibits a moderate increase across models, whereas MCQA accuracy improves by a substantially larger margin at comparable F1 levels. This indicates that baseline models already achieve reasonable *recognition* capability, but their low performance under our benchmark probing stems from *retrieval* as the primary bottleneck. Consequently, we show that the performance gains achieved by CoViP are driven primarily by improved retrieval, specifically more effective integration of implicit personal cues in captions through contextualized reasoning, rather than by improvements in recognition itself.

**Human evaluation supports the fidelity of CapEval-QAs.** We conduct a human evaluation to assess whether our evaluation protocol, CapEval-QAs, better aligns with human preferences. We compare against three baselines and report the results in Fig. 5. Captions preferred by CapEval-QAs consistently obtain higher win-or-tie rates across both human evaluation criteria, suggesting that our QA-based metric captures aspects of caption quality that are better aligned with human judgment. See further details in Appendix H.

Specifically, (i) *Context Groundedness*: in terms of contextual grounding, we demonstrate a clear preference advantage

*Table 4.* Performance improvements of CoViP over the baseline across multi-image benchmarks.

| Benchmark | Qwen3-VL-8B | CoViP | Δ (CoViP – Qwen3) |
|---|---|---|---|
| MM-NIAH (Wang et al., 2024c) | 86.7% | 88.0% | +1.3% |
| MMNeedle (Wang et al., 2025a) | 47.3% | 51.3% | +4.0% |
| MuirBench (Wang et al., 2024a) | 79.8% | 81.8% | +2.0% |
| MMIU (Meng et al., 2024) | 48.3% | 49.8% | +1.5% |
| Average | **65.5%** | **67.7%** | **+2.2%** |

over all baselines. (ii) *New Image Description*: regarding the quality of describing the new query image, CoViP performs on par with Qwen3-VL-8B, with no significant performance gap observed relative to other baselines. Importantly, these results indicate that CoViP does not overemphasize contextual recall at the expense of accurately describing the new image, further supporting its validity as a balanced framework for contextualized visual personalization.

**CoViP improves generalization on multi-image benchmarks.** We conduct additional experiments to assess whether CoViP improves performance on established multi-image benchmarks that evaluate visual grounding and cross-image identity matching. As shown in Table 4, CoViP consistently improves over the base model across all evaluated benchmarks. Further discussion on multi-image benchmarks and caption quality preservation is provided in Appendices F and G, respectively.

# 7. Conclusions and Limitations

**Summary of findings.** We introduce a new paradigm of contextualized visual personalization, defining it as grounding visual understanding in user-specific past visual experiences provided in the model's context and using this grounding to guide personalized outputs. Our framework, CoViP, enables holistic personalization through post-training on a novel, challenging personalized image captioning benchmark, along with caption-augmented generation (CAG). Extended studies on our diagnostic tasks demonstrate that CoViP serves as a foundational stage for contextualized visual personalization, enhancing both the robustness and generalizability of existing VLMs.

**Limitations.** Our benchmark relies on synthetic dialogues and generated images, which may contain factual inconsistencies or visual artifacts. Although we applied rigorous model-based quality filtering for benchmark construction, additional human verification could further improve the factuality of the dialogues and the visual fidelity of the images.

**Future directions.** A key next step is to develop personalization benchmarks grounded in omnimodal (Oh et al., 2026) and real-world user signals, such as shopping histories, conversational voice recordings, and long-term user–model interaction logs, enabling more practical evaluation across diverse use cases.

## Impact Statement

This work proposes a new paradigm for contextualized visual personalization in VLMs, enabling responses grounded in user-specific visual experiences and historical context. By introducing a challenging benchmark and a principled post-training framework, CoViP achieves reliable and robust performance across downstream personalization tasks, offering potential benefits for real-world applications such as proactive personal AI assistants and context-aware AI systems. However, enhanced personalization necessitates important ethical considerations, as leveraging personal visual histories may risk exposing sensitive information if deployed without adequate safeguards. While our benchmark and post-training framework are intended for controlled research settings, future work should investigate privacy-preserving mechanisms to ensure that advances in contextualized visual personalization are deployed responsibly and for the benefit of users.

## Acknowledgements

The authors gratefully acknowledge the support from the NVIDIA Academic Grant Program. This work was supported by the Institute of Information & Communications Technology Planning & Evaluation (IITP) grants funded by the Korea government (MSIT) [NO.RS-2021-II211343, Artificial Intelligence Graduate School Program (Seoul National University); No.2022-0-00959, RS-2022-II220959], by the National Research Foundation of Korea (NRF) grant [No.2022R1A3B1077720, 2022R1A5A7083908], BK21 FOUR Program of the Education and Research Program for Future ICT Pioneers, Seoul National University in 2026, and by the Samsung Electronics Co., Ltd [IO250520-12926-01].

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

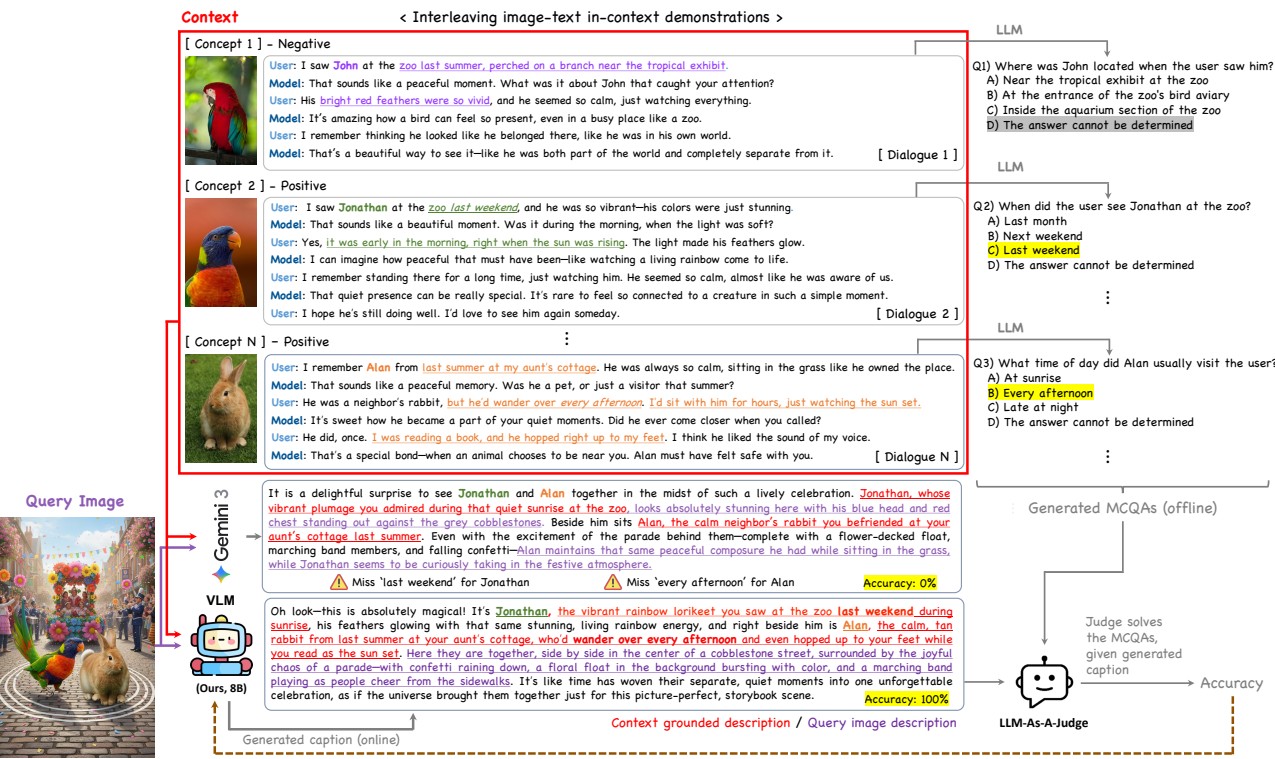

*Figure S.1.* Overview of our personalized image captioning framework. Given a query image, a VLM generates a personalized caption by referencing only the positive concept images and their associated dialogues within the context. An external LLM then evaluates whether the generated caption alone provides sufficient information to correctly answer a set of MCQs, using accuracy as the evaluation metric. During RL-based post-training, this accuracy is further leveraged as a VR signal.

## A. Related Works

**Contextualized personalization in LLMs.** Contextualized personalization in large language models (LLMs) refers to the ability of a model to generate responses aligned with a user's experiences contained in the prompt context (Liu et al., 2025a; Bai et al., 2022). Building on this foundation, a growing body of work has explored settings in which user profiles (Wang et al., 2024b), past dialogues (Baek et al., 2024), and previously shown content (Salemi et al., 2024b; Mok et al., 2025; Kim et al., 2025b) are incorporated into the context, allowing LLMs to produce personalized responses. More recently, studies have investigated personalization frameworks that retrieve past content relevant to the current user query, construct an appropriate memory, and condition the LLM's response on this memory (Hu et al., 2025; Salemi et al., 2024a; Zhou et al., 2025; Xiao et al., 2025). However, LLMs still frequently fail to effectively ground their outputs in the provided contextual information, and addressing this limitation remains an open challenge in existing research (Liu et al., 2024; Yu et al., 2025). In this work, we transfer the paradigm of contextualized personalization, actively studied in LLMs, to the setting of visual personalization in VLMs and conduct a systematic exploration of this problem.

## B. Detailed Description of Evaluation Protocol in Personalized Image Captioning Benchmark

We adopt a multiple-choice question-answering (MCQA) formulation to evaluate caption-level personalization, as it enables precise, interpretable measurement of contextual grounding. Unlike open-ended caption evaluation, which is often sensitive to surface-level lexical variation, MCQA enables us to directly assess whether a generated caption provides sufficient evidence to support specific contextual inferences.

In particular, the inclusion of distractor options and an explicit "cannot be determined" choice enables us to distinguish between three behaviors: (i) correctly utilizing relevant contextual information, (ii) hallucinating unsupported details, and (iii) appropriately abstaining when evidence is insufficient. This design is especially well-suited for evaluating personalization, where the key challenge lies not in visual recognition itself, but in selectively retrieving and applying user-specific contextual

---

**Algorithm 1** Algorithm for caption-based MCQA Probing of CoVIP

---

**Require:** Personalization model $p_\theta$, judge LLM $\mathcal{J}$; dataset $\mathcal{D} = \{(x_i, c_i, \mathrm{QA}_i)\}_{i=1}^N$.

  1:    Here, $\pi_\theta$ denotes the captioning policy; $x_i$ is the query image; $c_i$ is the interleaved context; $\mathrm{QA}_i$ is a concept-indexed QA bank; and $g_i$ denotes the positive concept set.

**Ensure:** Per-split metrics: $\mathrm{Acc}^+$ (Positive accuracy), $\mathrm{Acc}^-$ (Negative accuracy), and verifiable reward $r_{\mathrm{caps}}$.

  2:  **for** $i \leftarrow 1$ to $N$ **do**
  3:      Sample personalized caption $s_i \sim \pi_\theta(\cdot \mid x_i, c_i)$.
  4:      $\mathcal{Q}_i^+ \leftarrow \bigcup_{k \in g_i} \mathrm{QA}_i[\texttt{concept\_}k]$                                      {Positive QAs}
  5:      $\mathcal{Q}_i^- \leftarrow \bigcup_{k \notin g_i} \mathrm{QA}_i[\texttt{concept\_}k]$                                    {Negative QAs}
  6:      **for** each $q \in \mathcal{Q}_i^+ \cup \mathcal{Q}_i^-$ **do**
  7:         Sample outputs of $\mathcal{J}$ following the evaluation prompt $\psi(s_i, q)$:
  8:         $o(q) \leftarrow \mathcal{J}(\psi(s_i, q))$.
  9:      **end for**
10:      Scoring with accuracy metrics:
11:      $\mathrm{Acc}_i^+ \leftarrow \frac{1}{|\mathcal{Q}_i^+|} \sum_{q \in \mathcal{Q}_i^+} \mathbb{I}[o(q) = a^\star(q)]$, where $a^\star(q)$ are GT answer for $q$.
12:      $\mathrm{Acc}_i^- \leftarrow \frac{1}{|\mathcal{Q}_i^-|} \sum_{q \in \mathcal{Q}_i^-} \mathbb{I}[o(q) = D]$.
13:      (Assign VRs only if on-policy RL training)
14:      $\sigma_i^- \leftarrow \frac{1}{|\mathcal{Q}_i^-|} \sum_{q \in \mathcal{Q}_i^-} \mathbb{I}[o(q) \neq D]$
15:      Calculate the sample-level VR:
16:      $r_{\mathrm{caps}}(i) \leftarrow \mathrm{Acc}_i^+ - \alpha \sigma_i^-$, where $\alpha$ is a coefficient that controls the strength of the penalty.
17:  **end for**
18:  **Aggregate** $\mathrm{Acc}^+ \leftarrow \frac{1}{N_{pos}} \sum_i \mathrm{Acc}_i^+$,   $\mathrm{Acc}^- \leftarrow \frac{1}{N_{neg}} \sum_i \mathrm{Acc}_i^-$, where $N = N_{pos} + N_{neg}$

---

knowledge. As a result, MCQA probing provides a reliable and scalable proxy for measuring how effectively a model internalizes and utilizes personalized context in caption generation.

For each pre-generated dialogue described in Section 3.2.1, we construct a set of discriminative multiple-choice question answering (MCQA) items using an external large language model. These questions are designed to probe whether a generated caption accurately reflects the contextual information embedded in the dialogue, focusing on attributes such as temporal cues, personalized descriptions, and user-specific experiences.

Concretely, each dialogue yields three MCQA samples. Each sample consists of four answer options: (i) one correct answer grounded in the dialogue context, (ii) two distractor options that are plausible but indistinguishable without access to the relevant contextual information, and (iii) an explicit "cannot be determined" option, which captures cases where the provided caption does not contain sufficient evidence to answer the question. This construction allows us to assess not only whether the caption captures relevant contextual details, but also whether it avoids hallucinating or over-generalizing beyond the available information.

To enable fine-grained evaluation of contextualized personalization, we explicitly distinguish between *positive* and *negative* concepts within each dialogue context. Formally, for each dialogue $d_i$, we construct a set of QA pairs $(q_{ik}, a_{ik})$ such that each pair is labeled as either positive or negative depending on whether it corresponds to a concept relevant to the query image. Positive QA pairs evaluate whether the caption successfully incorporates relevant contextual information, whereas negative QA pairs test whether the caption avoids incorrectly transferring unrelated contextual details. This distinction allows us to separately measure the model's ability to retrieve relevant personalized information and to suppress irrelevant or misleading context.

To quantitatively evaluate the degree of contextualized personalization in generated captions, we define two complementary metrics: *Positive Accuracy* and *Negative Accuracy*. These metrics assess whether a caption correctly incorporates relevant contextual information while avoiding irrelevant or misleading content.

Let $\mathcal{Q}^+$ and $\mathcal{Q}^-$ denote the sets of MCQA samples derived from positive and negative concepts, respectively. For each question–answer pair $(q_{ik}, a_{ik})$, a judge model $\mathcal{J}$ is provided with the generated caption $s$ and produces a predicted answer $\hat{a}_{ik} = \mathcal{J}(q_{ik}, s)$.

Positive Accuracy measures how well the generated caption captures contextually relevant information. It is defined as the proportion of positive-concept questions for which the judge model selects the correct answer:

$$\text{Acc}^+ = \frac{1}{|\mathcal{Q}^+|} \sum_{(q_{ik}, a_{ik}) \in \mathcal{Q}^+} \mathbb{I}\left[\mathcal{J}(q_{ik}, s) = a_{ik}\right], \tag{S.1}$$

where $\mathbb{I}[\cdot]$ denotes the indicator function. A higher Positive Accuracy indicates that the generated caption successfully encodes relevant personalized information grounded in the dialogue context.

Negative Accuracy evaluates whether the model avoids incorporating irrelevant contextual information. For questions derived from negative concepts, the correct behavior is to select the "cannot be determined" option, indicating that the caption does not contain sufficient evidence to answer the question. Formally, Negative Accuracy is defined as:

$$\text{Acc}^- = \frac{1}{|\mathcal{Q}^-|} \sum_{(q_{ik}, a_{ik}) \in \mathcal{Q}^-} \mathbb{I}\left[\mathcal{J}(q_{ik}, s) = \texttt{cannot\_be\_determined}\right]. \tag{S.2}$$

While Positive Accuracy reflects the model's ability to retrieve and express relevant personalized information, Negative Accuracy captures its robustness against hallucination and over-generalization. Together, these metrics provide a balanced and interpretable evaluation of caption-level personalization, measuring both selective recall and selective omission of contextual information.

## C. Implementation Details of the Post-training Pipeline

**Post-training details.** To mitigate training instability caused by small batch sizes, we employ gradient accumulation with 2 steps for both GRPO and DrGRPO, and 1 step for GSPO. To prevent excessively verbose generations, the maximum completion length is capped at 511-tokens. Our implementation builds on an open-source codebase.[2] For training, we apply LoRA with rank 64 and scaling factor (alpha) 128, and sample 4 rollouts per prompt to compute group-level advantages. We adopt `Qwen3-VL-Instruct-8B` as the backbone VLM, which supports multimodal processing of images and text and exhibits strong instruction-following capabilities. This choice is motivated by its open-source availability and its ability to handle multi-image and long-context inputs, which are essential for contextualized personalization. We use this model for on-policy RL post-training.

**Optimization strategy for CoViP.** GSPO (Zheng et al., 2025) is an on-policy RL algorithm that (i) samples a group of $G$ completions per input and (ii) performs sequence-level off-policy correction and clipping, aligning the optimization unit with the sequence-level reward. For each $(x, c)$, we roll out a group $\{s_i\}_{i=1}^G \sim \pi_{\theta_{\text{old}}}(\cdot \mid x, c)$, compute VR $r_i = r(s_i, x, c)$ from Eq. (1) of our main paper, and define the group-normalized advantage

$$\hat{A}_i = \frac{r_i - \text{mean}(\{r_j\}_{j=1}^G)}{\text{std}(\{r_j\}_{j=1}^G)}. \tag{S.3}$$

It is worth noting that GSPO uses a sequence-likelihood importance ratio $\nu_i(\theta)$ with length normalization:

$$\begin{aligned}
\nu_i(\theta) &= \left(\frac{\pi_\theta(s_i \mid x, c)}{\pi_{\theta_{\text{old}}}(s_i \mid x, c)}\right)^{\frac{1}{|s_i|}} \\
&= \exp\left(\frac{1}{|s_i|} \sum_{t=1}^{|s_i|} \log \frac{\pi_\theta(s_{i,t} \mid x, c, s_{i,<t})}{\pi_{\theta_{\text{old}}}(s_{i,t} \mid x, c, s_{i,<t})}\right).
\end{aligned} \tag{S.4}$$

and optimizes the clipped surrogate objective at the sequence level:

$$J(\theta) = \mathbb{E}\left[\frac{1}{G} \sum_{i=1}^G \min\left(\nu_i(\theta)\, \hat{A}_i,\ \text{clip}(\nu_i(\theta), \epsilon_1, \epsilon_2)\, \hat{A}_i\right)\right], \tag{S.5}$$

---

[2] `https://github.com/om-ai-lab/VLM-R1`

where $\epsilon_1 = 1 - \epsilon$, $\epsilon_2 = 1 + \epsilon$. Unlike token-level importance weighting (Shao et al., 2024), GSPO assigns a single clipped weight to each sequence. This approach aligns with our reward design, which evaluates the entire caption as a single unit, thereby enhancing training stability—particularly for long-form generation. Furthermore, length-normalization of the ratio mitigates variance and ensures that importance weights remain within a comparable numerical range across varying caption lengths.

**Degeneration filtering.** As noted in Eq. (6) in our main paper, to prevent reward hacking via degenerate repetition, we apply a hard penalty to repetitive or overly long generations when computing $r_{\text{caps}}$. Let $\rho_{\text{sent}}$, $\rho_{n\text{-gram}}$ (with $n{=}5$), and $\rho_{\text{chunk}}(L)$ (with $L \in \mathcal{L}$ and $\mathcal{L} = \{10, 20, 30\}$ words) denote sentence-level, $n$-gram-level, and chunk-level duplication ratios, respectively. Formally, we define a repetition-based degeneration indicator as

$$\delta(y) = \mathbb{I}\Big[\rho_{\text{sent}} \geq \tau_s \ \vee \ \rho_{n\text{-gram}} \geq \tau_n \ \vee \ \max_{L \in \mathcal{L}} \rho_{\text{chunk}}(L) \geq \tau_c\Big], \tag{S.6}$$

where $\tau_s{=}0.3$, $\tau_n{=}0.3$, and $\tau_c{=}0.2$. To further discourage verbose generations, we also impose a length constraint. Let $|y|$ denote the tokenizer encoding length and $l$ be a fixed threshold. We then define the overall degeneration filtering predicate as

$$\text{R}(y) = \big(\delta(y) = 1\big) \ \vee \ \big(|y| > l\big). \tag{S.7}$$

## D. Experimental Configurations

**Used VLMs** For generating dialogues, we employ the VLM `Qwen/Qwen3-VL-30B-A3B-Instruct-FP8`. To generate MCQA pairs from these dialogues and to calculate accuracy-based VR via an *LLM-as-a-judge*, we use the `Qwen/Qwen3-30B-A3B-Instruct-2507-FP8` LLM. Across all experiments, the decoding temperature is fixed to `0.0` to ensure deterministic generation.

**Design of Diagnostic Personalization Tasks** We design a suite of diagnostic tasks to evaluate *implicit visual personalization* in VLMs. These diagnostics are guided by a core principle: the model must *recognize* the individual depicted in a query image using visual evidence and *retrieve* relevant personalized experiences from contextual memory, while shortcut behaviors that bypass visual grounding are explicitly disallowed. Collectively, these tasks are designed to quantitatively assess a model's ability to perform personalized reasoning grounded in visual identity. An overview of the diagnostic tasks is illustrated in Figure 3.

To construct the diagnostic tasks, we require multiple images of the same individual to enable identity-level reasoning across contexts. We therefore sample 50 individuals from the MMPB dataset (Kim et al., 2025a), collecting 4 images per person, resulting in a total of 200 human images. For each individual, one image is used as the *query image*, while the remaining three images are used within contextual dialogues.

For each context, we generate a total of 10 visual-textual dialogues. Among them, 3 dialogues correspond to the same individual as the query image, while the remaining 7 correspond to different individuals randomly sampled from the other 49 identities. Each individual serves as the query subject in 10 different contexts, yielding a total of 500 contexts.

Each dialogue is automatically generated using a VLM-based generator. For the three dialogues associated with the query individual, the generator is provided with a randomly sampled name, location, and date to construct personalized experiences. The prompt used for dialogue generation is provided in Table S.18.

Across all diagnostic tasks, the model is required to: (1) recognize the identity of the person in the query image, (2) retrieve all relevant dialogues associated with that individual from the context, and (3) perform temporal reasoning to identify the most recent interaction.

This setup prevents shortcut solutions such as only-text matching or partial context retrieval, as correct prediction requires joint visual recognition and memory-based reasoning over multiple dialogue instances.

1. **Last-Seen Detection (LSD).** In the LSD task, each dialogue contains an explicit reference to when and where the user encountered the individual, e.g., *"I still remember the day I saw John at Lake Francesborough on 2025-09-09."*

   Given a new query image, the user asks: *"Where did I last see the person in this image?"*

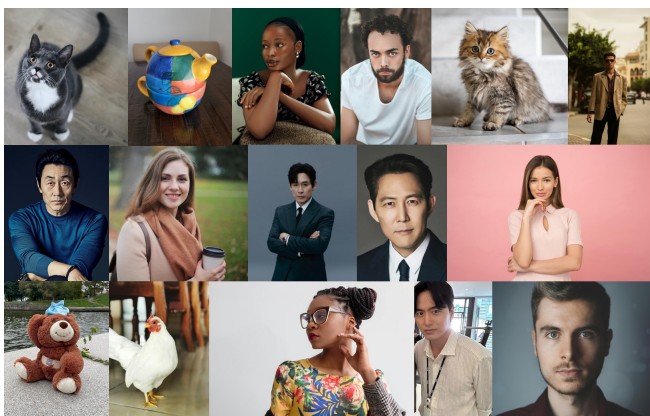

*Figure S.2.* Used real images to construct our database.

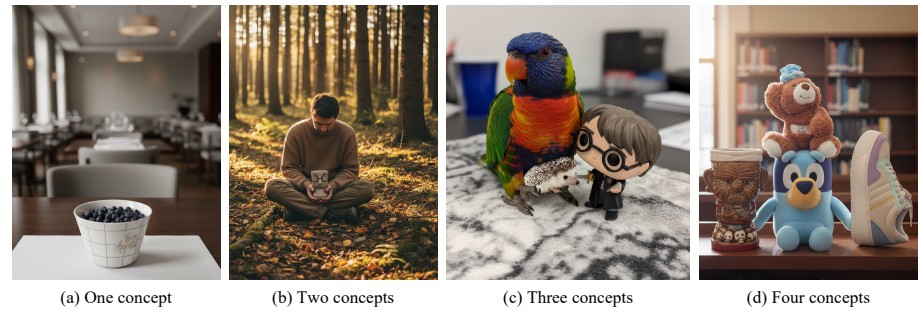

| (a) One concept | (b) Two concepts | (c) Three concepts | (d) Four concepts |

*Figure S.3.* Visualization of query images generated with varying numbers of concept images.

To answer correctly, the model must identify the person, retrieve all associated dialogues, compare their timestamps, and extract the location from the most recent one. Performance is measured using word-level F1 between the model output and the ground-truth location.

2. **Last-Action Recall (LAR).** LAR extends LSD by requiring recall of a finer-grained personal action rather than a location. For each context, we append an additional user utterance to the *last-seen dialogue* of the query individual, describing a specific action, e.g., *"Oh wait, I need to go to the post office to return this package."*

   The action is randomly sampled from a predefined candidate set (Table S.20) and injected into all 500 contexts. Given a query image, the user asks: *"What was I doing the last time I mentioned the person in this image?"*

   The model must identify the correct individual, locate the most recent dialogue, and retrieve the embedded action. Unlike LSD, which focuses on factual recall, LAR evaluates the model's ability to retrieve fine-grained, episode-level user states. We evaluate performance using an LLM-based judge that determines whether the generated response semantically matches the ground-truth action (see Table S.21).

3. **Instruction-Triggered Recall (ITR).** ITR evaluates a more proactive form of personalization. In this task, the last-seen dialogue includes an instruction of the form: *"If this person ever shows up again, remind me by saying the keyword* SKS.*"*

   At inference time, the user asks a generic question such as: *"Where did I last see the person in this image?"* without explicitly mentioning the keyword.

   To succeed, the model must not only recognize the individual and retrieve the relevant memory, but also proactively incorporate the specified trigger keyword (SKS) into its response. Unlike LSD and LAR, which evaluate reactive personalization, ITR assesses whether the model can perform *proactive personalization* conditioned on visual recognition and prior instructions.

   We evaluate ITR using the trigger success rate, defined as the proportion of responses that include the keyword SKS.

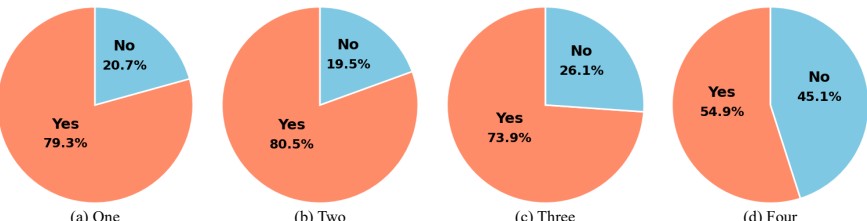

*Figure S.4.* Quality filtering results obtained with `Gemini-2.5 Flash`. 'Yes' denotes the proportion of retained query images after filtering.

*Table S.1.* Comparison of synthetic and real-world multimodal personalization data.

| Data | Personal Data / Potential Bias | Data Acquisition | Scenarios | Modalities | Expense | Evaluation |
|---|---|---|---|---|---|---|
| Synthetic (this paper) | Homogeneous context (evaluation-oriented) | Easy (Gemini-3.0-Pro + quality filtering) | Conversation, experience-oriented | Image, Text | Relatively cheap (API cost) | Easy (MCQA) |
| Real-world | Heterogeneous context (noisy, missing modalities) | Very hard (manual annotation, privacy constraints) | Shopping history, medical logs, e-mails | Image, Text, Video, Audio | Highly expensive / mostly unavailable | Very hard (noisy, multi-hop, no GT) |

**Procedures for Benchmark Construction**   Prior works adopt various data acquisition strategies to construct evaluation sets. For example, RAP leverages frame-level sampling from YouTube videos followed by automated caption generation, while RePIC curates images from movie teasers and award ceremonies where multiple celebrities co-occur. Despite these efforts, collecting diverse real-world evaluation datasets that simultaneously exhibit varied backgrounds and offer fine-grained, controllable visual granularity for each concept remains a major bottleneck. To address this limitation, we construct a benchmark composed of realistically generated images. Specifically, we built a database of 188 images, consisting of 123 human images, 34 object images, and 31 pet images. Each image corresponds to a unique identity, ensuring no identity overlap across the dataset. The database is carefully curated to support fine-grained visual personalization.

- For the human category, images were collected from a combination of open-source datasets, including Yo'LLaVA (Nguyen et al., 2024) and CelebA (Liu et al., 2015), as well as through web crawling from royalty-free image platforms such as Unsplash and Pexels.

- For the object and pet categories, we utilized subset images from existing personalization benchmarks, including MyVLM (Alaluf et al., 2024), Custom101 (Kumari et al., 2023), RAP-LLaVA (Hao et al., 2025), and DreamBooth (Ruiz et al., 2023).

**Benchmark quality filtering results.**   As shown in Figure S.4, images synthesized with one or two concepts generally align well with the prompts, whereas the failure rate increases noticeably starting from three concepts. Specifically, for four concepts, nearly 45% of the synthesized images fail to faithfully reflect the specified prompts. Accordingly, more stringent quality filtering is applied for samples involving three or four concepts. Note that visualizations of representative samples are presented in Figure S.3.

## E. Further Limitations

**Data Transition Directions**   As shown in Table S.1, actively incorporating real-world data at evaluation time remains challenging due to privacy concerns (*e.g.*, medical logs) and the lack of reliable human annotations, a limitation that persists across current open-source datasets and benchmarks. While we propose a benchmark comprising both synthetic and real-world data for contextualized visual personalization, we acknowledge that potential biases exist and that transitioning toward more diverse, real-world contexts is a necessary next step. Such a transition also introduces additional challenges, as real-world data increasingly requires multi-hop reasoning. For instance, inferring a user's implicit preference requires

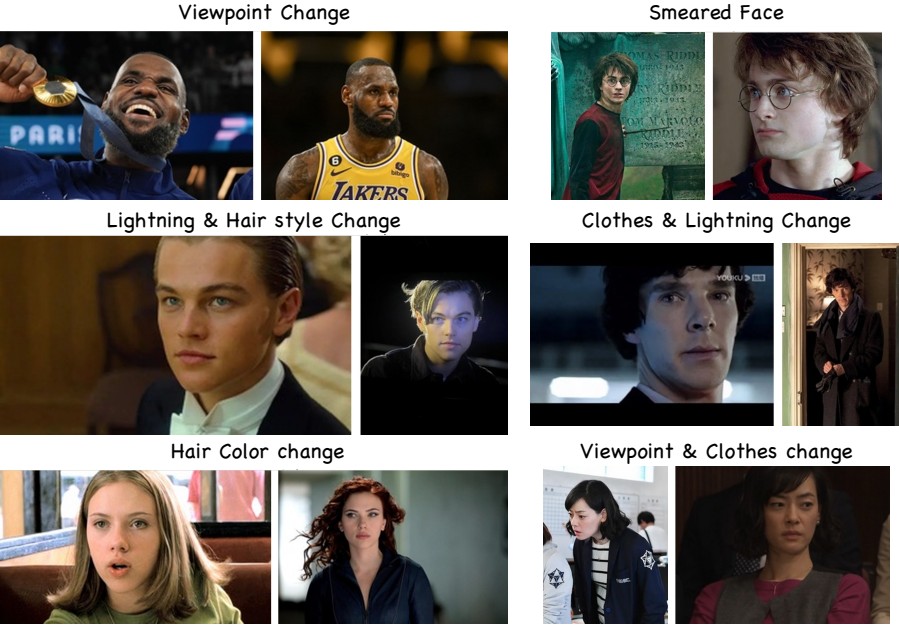

*Figure S.5.* Representative real images used exclusively for downstream diagnostic evaluation (LSD, LAR, and ITR), not seen during training. The examples illustrate varying granularities of perceptual difference between context and query images.

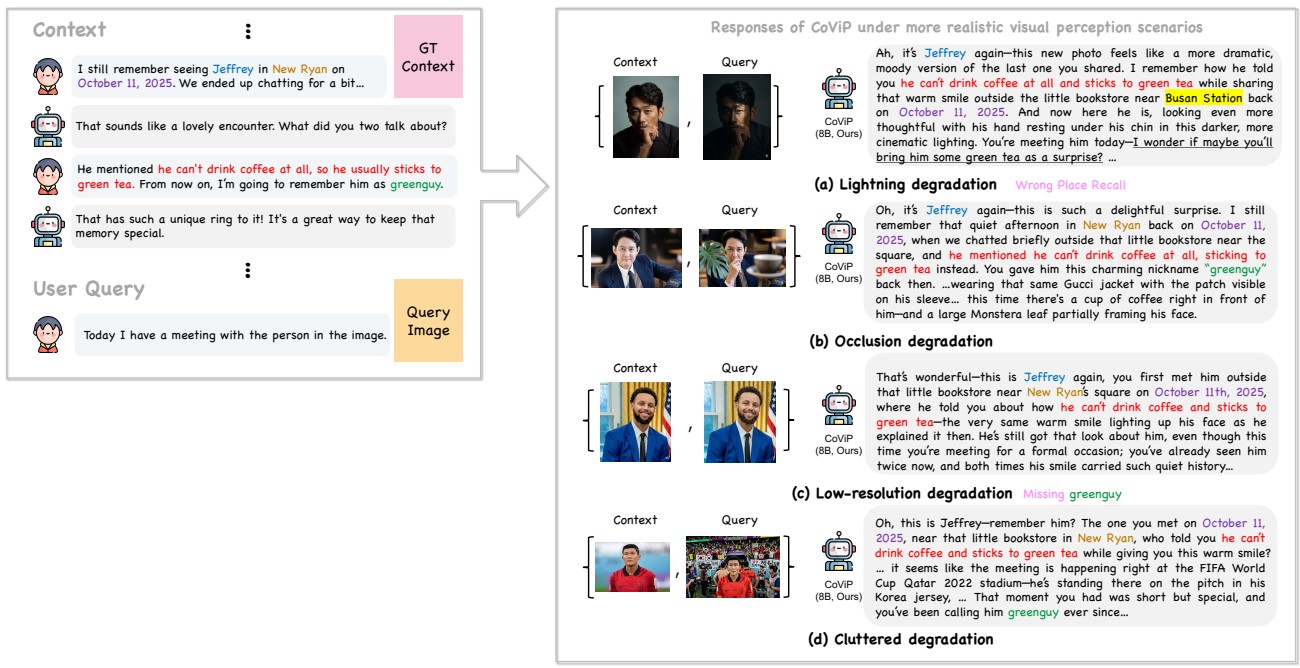

*Figure S.6.* Qualitative examples of CoViP inference under imperfect visual recognition in realistic real-world environments, generated by Gemini-3.0-Pro.

first identifying relevant past data across multiple heterogeneous memory sources, and then reasoning about what that information implies for the current query. Given these compounding challenges in real-world settings, we deliberately chose to simplify the problem setting using synthetic data in order to isolate and study the core capability of contextualized visual personalization in a controlled manner.

While our current benchmark relies primarily on synthetic data, we believe that, as the first work to formally define and study contextualized visual personalization, our contribution lies in establishing a rigorous and diagnostic foundation upon

*Table S.2.* Ablation results of different algorithmic variants for on-policy RL post-training.

| Models | 1-Concept | | 2-Concepts | | 3-Concepts | | 4-Concepts | |
|---|---|---|---|---|---|---|---|---|
| Accuracy (%) | Pos Acc | Neg Acc | Pos Acc | Neg Acc | Pos Acc | Neg Acc | Pos Acc | Neg Acc |
| Qwen3-GRPO (Shao et al., 2024) | 67.1 | **95.5** | 53.3 | **95.5** | 47.6 | **96.1** | 41.4 | 95.2 |
| Qwen3-Dr.GRPO (Liu et al., 2025b) | 68.3 | 94.2 | 51.6 | 94.7 | 48.6 | 95.1 | 43.3 | **95.9** |
| Qwen3-GSPO (Zheng et al., 2025) | **72.2** | 94.7 | **53.9** | 94.8 | **51.3** | 95.4 | **45.7** | 95.3 |

which future work can build. We hope that this formalization of the problem will guide subsequent research toward better alignment with real-world data distributions.

**Textual robustness of CoViP**    Real-world in-context memory would likely be less structured, potentially conflicting, and considerably noisier or more ambiguous. We had also considered incorporating more heterogeneous context during dataset construction. However, doing so often introduced unintended textual hints within specific dialogues, making it difficult to prevent shortcut learning while maintaining a clean and diagnostic evaluation setup. Although we intentionally prioritized constructing a shortcut-free benchmark, we conjecture that our model may fail under hard conditions, though quantifying the extent remains non-trivial.

**Visual robustness of CoViP**    We acknowledge that the visual perception capability covered by our benchmark is limited to relatively fine-grained perception within the context/query setup, and does not extend to open-ended real-world robustness issues. Although we did not separately cluster or report these cases in the main paper, our downstream diagnostic evaluations do include visually challenging cases, and we provide additional examples in Fig. S.5. In addition, we include qualitative examples in Fig. S.6 regarding imperfect visual perception.

## F. Additional Analysis

**Analysis of training dynamics**    Table S.2 reports an ablation over on-policy training algorithms, including GRPO, DrGRPO, and GSPO. Notably, GSPO consistently achieves the best performance across all concepts; therefore, we adopt GSPO as our default on-policy RL post-training algorithm.

We further interpret these evaluation results through the lens of training-time reward trajectories. In Figure S.7(a), naively applying GSPO leads to a gradual decline in accumulated reward as training proceeds, especially in later stages. This suggests that without degeneration filtering, optimization can drift toward misleading behaviors that degrade the personalization performance. Importantly, introducing our degeneration filtering further stabilizes training: completion length gradually converges while reward improves. This indicates that the VLM learns to selectively include only contextually relevant information without introducing verbosity, producing concise yet context-grounded captions that better explain the query.

In Figure S.7(b), GRPO converges more slowly due to KL-divergence regularization, whereas DrGRPO and GSPO, which do not use KL regularization, exhibit faster convergence. However, DrGRPO is notably unstable and tends to diverge early in training. In contrast, our GSPO-based post-training remains stable and achieves substantially higher rewards, indicating a favorable trade-off between optimization efficiency and training stability.

Finally, Figure S.7(c) highlights the role of prompt design in training stability. When we randomly alternate prompt roles (*i.e.*, switching between user and system prompts) instead of using a fixed captioning prompt, the completion length rapidly diverges. We posit that this behavior arises because the VLM loses instruction-following fidelity and thus tends to incorporate information from both relevant and irrelevant dialogues indiscriminately, leading to excessively verbose outputs.

**Analysis of $\text{Acc}^-$ results in Table 2**    As shown in Table 2, the VLM post-trained with CoViP achieves a substantially higher $\text{Acc}^+$ compared to the baseline, while exhibiting a slight decrease in $\text{Acc}^-$. We argue that this decrease does not indicate a degradation in the model's actual performance. Rather, it arises as a consequence of improved contextual understanding and retrieval behavior.

Specifically, the baseline model often succeeds in loosely matching the query image to a relevant dialogue based on high-level visual concepts, yet fails to retrieve detailed user-specific information from that dialogue. This limitation is reflected in its relatively low $\text{Acc}^+$. As a result, even when the model misrecognizes the underlying concept, it frequently

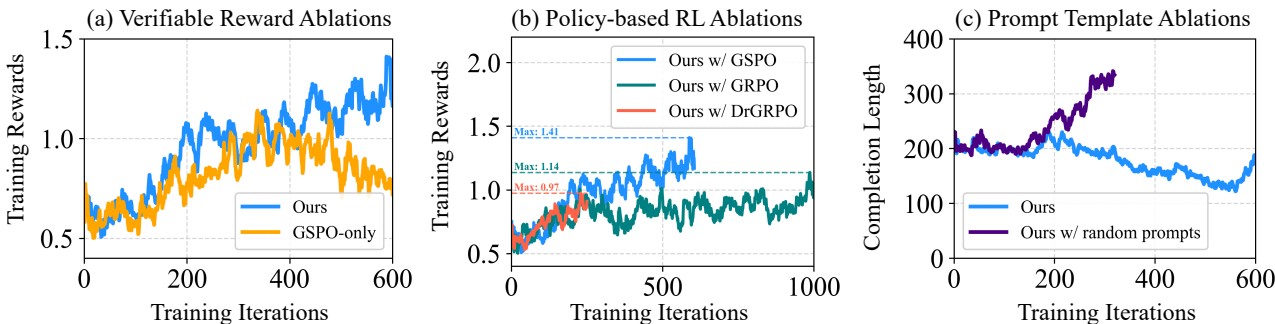

*Figure S.7.* Reward trajectory ablations on VR designs, on-policy RL algorithms, and prompt templates during post-training.

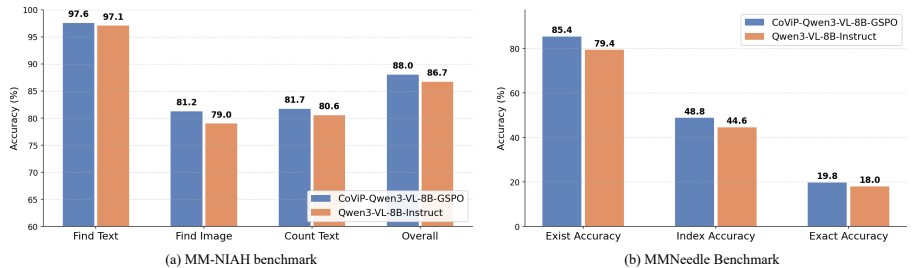

*Figure S.8.* Results on MM-NIAH (∼2.8K samples) and MMNeedle (∼1K subset, 4x4 grid), which evaluate long-context multimodal retrieval and needle localization in multi-image documents. CoViP consistently outperforms Qwen3-VL across all subtasks, with overall gains of +1.3% and +4.0% on MM-NIAH and MMNeedle, respectively.

responds with "`cannot determine`" during MCQA evaluation, which is counted as a correct response under the $\text{Acc}^-$ metric, effectively producing false positives.

In contrast, after post-training via CoViP, the model becomes significantly more capable of retrieving fine-grained, experience-level details from the associated dialogue once a concept is recognized. Consequently, when the model mistakenly identifies a concept, it is now more likely to commit to a specific answer based on the retrieved information, leading to a lower $\text{Acc}^-$. Therefore, the observed decrease in $\text{Acc}^-$ does not reflect weaker reasoning ability; rather, it indicates that the model more actively incorporates personalized contextual knowledge into its predictions.

### Generalization to Other Benchmarks

In particular, CoViP improves performance on long-context visual retrieval benchmarks in Fig. S.8 (MM-NIAH, MMNeedle) as well as multi-image visual grounding benchmarks in Figs. S.9 and S.10 (MuirBench, MMIU). These results suggest that our RLVR-based training objectives enhance not only personalized contextual grounding, but also the model's broader multi-image visual perception capabilities.

## G. Additional Results

**Experiments on other VLM personalization benchmarks.**   We report the performance of RePIC by reproducing the results with the Qwen3-VL baselines, and the corresponding results are reported. Since extensive hyperparameter optimization was not performed for this baseline, its performance may be slightly lower than the results reported in the original paper. Notably, as shown in Tables S.3 and S.4, CoViP shows comparable or even surpasses performances on the benchmark without relying on the VRs introduced in RePIC, demonstrating the effectiveness and generality of our approach.

**Ablations on VR designs.**   Table S.5 reports benchmark performance under different VR configurations. We further analyze (i) the contribution of each VR component and (ii) the effect of the number of concepts used during post-training.

1. **Necessity of joint supervision.** Training with only $r_{\text{vis}}$ (without $r_{\text{caps}}$) degrades performance, in some cases falling

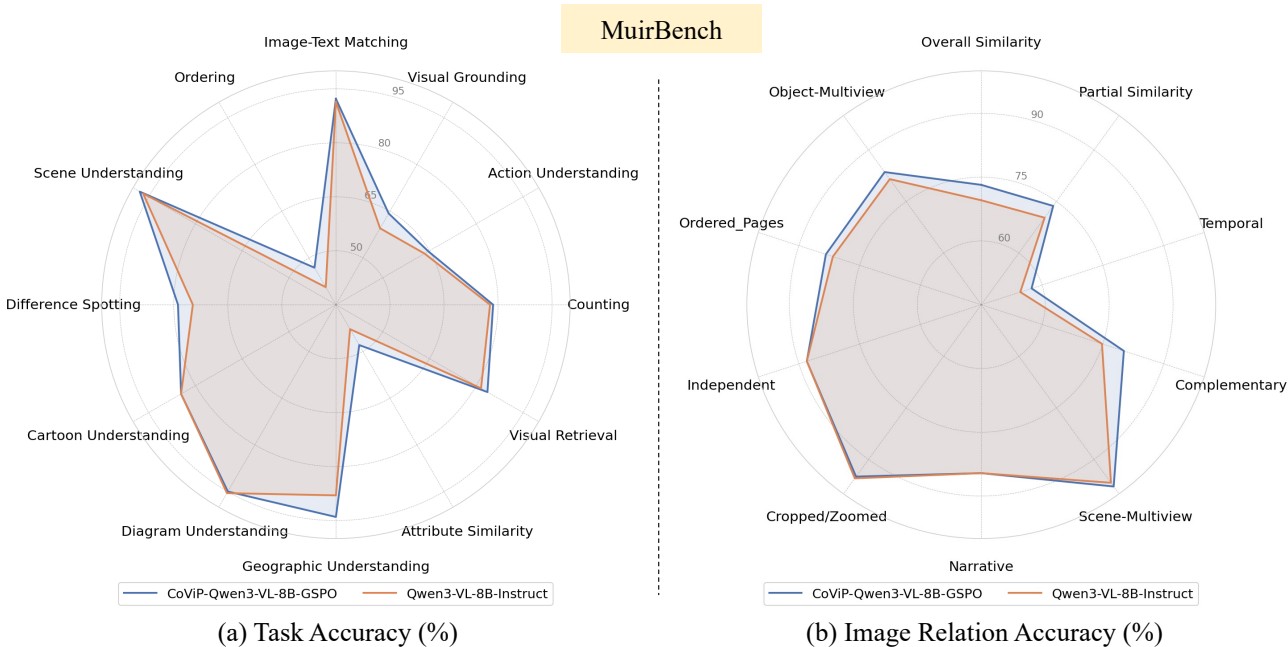

*Figure S.9.* Results on MuirBench (∼1.3K answerable MCQs), which evaluates robust multi-image understanding across 12 tasks and 10 inter-image relation categories. Especially, CoViP shows gains on visual perception tasks (e.g., Visual Grounding, Image-Text Matching, Partial Similarity) than Qwen3-VL.

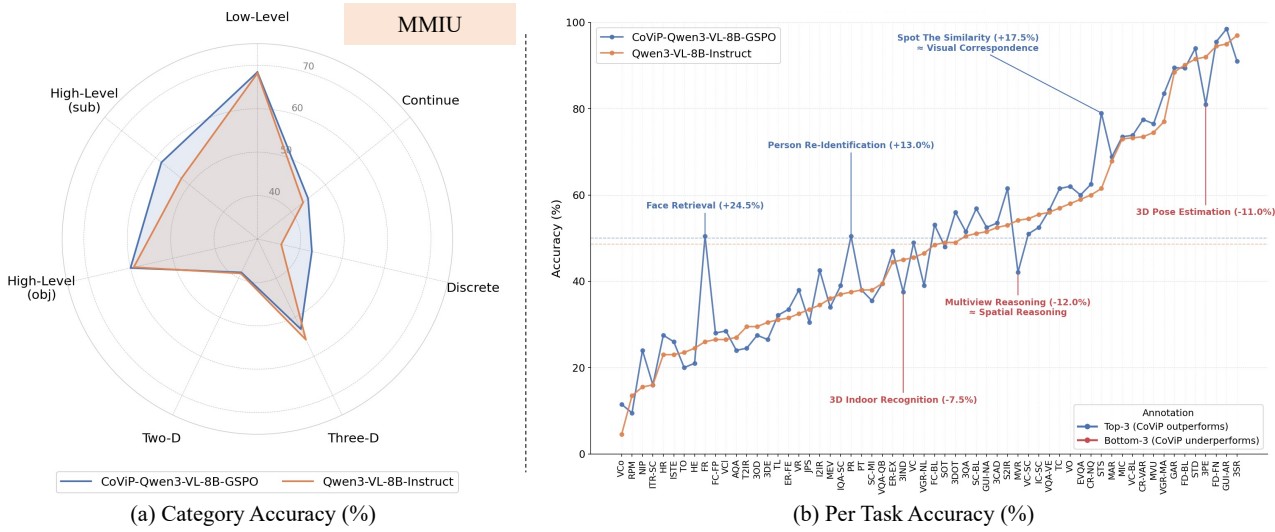

*Figure S.10.* Results on MMIU (11K MCQs across 52 tasks), which evaluates multi-image relational understanding spanning temporal, semantic, and spatial categories. CoViP achieves consistent category-level gains and the largest per-task improvements on identity- and appearance-based retrieval tasks directly aligned with its RLVR objective (Face Retrieval: +24.5%, Visual Correspondence: +17.5%, Person Re-ID: +13.0%), while underperforming on 3D spatial reasoning tasks outside its training scope.

below the `Qwen3-VL-8B` baseline, indicating that visual supervision alone is insufficient for personalized image captioning. Conversely, optimizing only $r_{caps}$ also yields consistently weaker results, suggesting retrieval signals without fine-grained visual supervision are inadequate.

2. **F1-based vision VR vs. binary consistency VR.** Replacing RePIC's object-consistency VR (OCT), which provides binary correctness feedback, with our set-based F1 VR $r_{vis}$ consistently improves performance on positive accuracy. This indicates that set-level supervision provides a denser and more robust learning signal for multi-concept perception.

*Table S.3.* Evaluation results of single-concept personalized grounding performance.

| Models | Backbone | MyVLM | | | Yo'LLaVA | | | DreamBooth | | |
|---|---|---|---|---|---|---|---|---|---|---|
| | | Pre. | Rec. | F1 | Pre. | Rec. | F1 | Pre. | Rec. | F1 |
| *Skip-Retrieval Setting* | | | | | | | | | | |
| Zero-shot | Qwen-2.5 VL | 100 | 56.8 | 72.4 | 99.6 | 83.2 | 90.7 | 96.0 | 76.6 | 85.2 |
| RAP | Qwen-2.5 VL | 100 | 98.8 | 99.4 | 100 | 99.8 | 99.8 | 100 | 100 | 100 |
| RePIC | Qwen-2.5 VL | 100 | 96.2 | 98.1 | 99.7 | 96.1 | 97.9 | 100 | 98.1 | 99.0 |
| Zero-shot | Qwen-3 VL | 100 | 54.7 | 70.7 | 98.0 | 73.9 | 84.2 | 98.3 | 73.4 | 84.1 |
| RAP | Qwen-3 VL | 100 | 99.7 | 99.9 | 100 | 76.6 | 86.7 | 100 | 99.4 | 99.7 |
| RePIC | Qwen-3 VL | 100 | 74.4 | 85.1 | 100 | 89.5 | 94.5 | 99.3 | 86.1 | 90.1 |
| CoViP | Qwen-3 VL | 100 | 76.8 | 86.9 | 100 | 85.3 | 92.1 | 100 | 86.7 | 92.9 |
| *Retrieval Setting* | | | | | | | | | | |
| Zero-shot | Qwen-2.5 VL | 91.5 | 50.6 | 65.2 | 77.4 | 42.3 | 55.2 | 95.2 | 75.3 | 84.1 |
| RAP | Qwen-2.5 VL | 95.5 | 87.9 | 91.6 | 79.2 | 75.1 | 76.2 | 98.7 | 94.3 | 96.4 |
| RePIC | Qwen-2.5 VL | 99.0 | 83.2 | 90.4 | 84.4 | 69.7 | 76.3 | 98.6 | 90.5 | 94.4 |
| Zero-shot | Qwen-3 VL | 90.4 | 49.7 | 64.1 | 74.7 | 60.4 | 66.8 | 98.3 | 72.7 | 82.9 |
| RAP | Qwen-3 VL | 93.6 | 90.6 | 92.1 | 76.6 | 61.0 | 67.9 | 98.0 | 94.3 | 96.1 |
| RePIC | Qwen-3 VL | 92.6 | 66.5 | 77.4 | 78.0 | 67.3 | 72.3 | 97.8 | 83.5 | 90.1 |
| CoViP | Qwen-3 VL | 95.2 | 70.0 | 80.7 | 77.8 | 68.5 | 72.8 | 98.5 | 84.8 | 91.2 |

*Table S.4.* Evaluation results of multi-concept personalized grounding performance.

| Models | Backbone | Skip-Retrieval | | | Retrieval | | |
|---|---|---|---|---|---|---|---|
| | | Pre. | Rec. | F1 | Pre. | Rec. | F1 |
| Zero-shot | Qwen-2.5 VL | 100 | 75.0 | 85.7 | 98.1 | 64.0 | 77.5 |
| RAP-Qwen | Qwen-2.5 VL | 100 | 82.9 | 90.7 | 100 | 73.2 | 84.5 |
| RePIC | Qwen-2.5 VL | 100 | 98.8 | 99.4 | 97.5 | 93.9 | 95.7 |
| Zero-shot | Qwen-3 VL | 100 | 82.9 | 90.7 | 100 | 80.5 | 89.2 |
| RAP-Qwen | Qwen-3 VL | 97.7 | 78.7 | 87.7 | 100 | 70.7 | 82.9 |
| RePIC | Qwen-3 VL | 100 | 95.1 | 97.5 | 99.3 | 86.0 | 92.2 |
| CoViP | Qwen-3 VL | 100 | 87.2 | 93.2 | 99.3 | 82.9 | 90.4 |

3. **Effect of increasing the number of positive concepts.** Varying the number of positive concepts included during post-training shows that using all four concepts yields the best performance.

**Comparing VRs used for object detection.** While prior works (Oh et al., 2025; Shen et al., 2025) use an Intersection over Union (IoU) score as a VR to strengthen localization capabilities, our proposed VR is fundamentally different from localization: it incentivizes the VLM to select only the relevant concepts, thereby balancing precision and recall to ensure discriminative visual grounding.

**Mitigating degeneration.** Finally, rather than naively applying GSPO, adding the proposed $n$-gram and chunking-based filtering further improves positive accuracy, especially in multi-concept settings. This indicates that discouraging repetitive or degenerate generations stabilizes training and steers optimization toward meaningful, context-grounded solutions.

**Ablations on different prompt roles.** In Table S.6, we conduct ablation experiments on personalized image captioning to analyze performance under different prompt configurations, including user prompts, system prompts, and settings without detailed task instructions. Here, randomized refers to a training setup that uses multiple prompt templates to reduce reliance on a single captioning prompt, while fixed denotes using the same prompt template during both training and inference.

We observe that even when a VLM is post-trained using only a fixed user prompt, it achieves slightly better performance than the randomized setting when evaluated with a system prompt at inference time. In contrast, removing detailed task descriptions from either the user or system prompt leads to a significant drop in performance.

These results demonstrate the effectiveness of the proposed captioning prompt as a user prompt. Importantly, the observed performance gains are not due to overfitting to a specific prompt template. Instead, the user prompt effectively encourages context-grounded responses, enabling CoViP to robustly include relevant contextual information even under changes in prompt roles. The user prompt used for image captioning is provided in Table S.13.

*Table S.5.* Ablation of VR components and the effect of the number of concepts used during training on test accuracy.

| Models
Accuracy (%) | 1-Concept
$Acc^+$ | $Acc^-$ | 2-Concepts
$Acc^+$ | $Acc^-$ | 3-Concepts
$Acc^+$ | $Acc^-$ | 4-Concepts
$Acc^+$ | $Acc^-$ |
|---|---|---|---|---|---|---|---|---|
| Qwen-3-VL-8B (Baseline) | 39.0 | 97.5 | 25.6 | 97.7 | 23.3 | 98.1 | 18.6 | 98.1 |
| *Ablations on Verifiable Rewards* | | | | | | | | |
| Qwen3-w/o $r_{caps}$ | 39.1 | 96.9 | 26.4 | 97.1 | 23.5 | 97.7 | 16.5 | 97.2 |
| Qwen3-w/o $r_{vis}$ | 61.0 | 95.0 | 44.9 | 95.1 | 39.8 | 95.7 | 34.5 | 96.7 |
| Qwen3-w/ OCT (Oh et al., 2025) | 77.0 | 91.9 | 63.1 | 92.1 | 58.3 | 91.8 | 59.1 | 83.4 |
| *Ablations on Training Concepts (Naive GSPO)* | | | | | | | | |
| Qwen3-upto 1-concept | 63.7 | 94.4 | 45.4 | 95.4 | 41.4 | 96.1 | 36.0 | 96.6 |
| Qwen3-upto 2-concepts | 68.4 | 94.7 | 53.2 | 94.9 | 50.6 | 95.2 | 46.3 | 95.5 |
| Qwen3-upto 3-concepts | 65.4 | 92.4 | 54.4 | 93.6 | 51.1 | 94.9 | 45.7 | 95.4 |
| Qwen3-Full | 72.2 | 94.7 | 53.9 | 94.8 | 51.3 | 95.4 | 45.7 | 95.3 |
| *Training with All of Proposed VRs & Degeneration Filtering* | | | | | | | | |
| Qwen3-VL-8B + CoViP | **77.4** | 94.8 | **68.4** | 94.1 | **65.2** | 94.8 | **59.7** | 92.8 |

*Table S.6.* Evaluation of generalization under variations of prompt templates (roles) applied during post-training.

| Models
Accuracy (%) | 1-Concept
$Acc^+$ | $Acc^-$ | 2-Concepts
$Acc^+$ | $Acc^-$ | 3-Concepts
$Acc^+$ | $Acc^-$ | 4-Concepts
$Acc^+$ | $Acc^-$ |
|---|---|---|---|---|---|---|---|---|
| *w/ User Prompt* | | | | | | | | |
| Randomize | 71.7 | 92.2 | 58.8 | 91.7 | 53.5 | 93.7 | 50.0 | 94.0 |
| Fixed | **77.4** | 94.8 | **68.4** | 94.1 | **65.2** | 94.8 | **59.7** | 92.8 |
| Zero-Shot | 39.0 | 97.5 | 25.6 | 97.7 | 23.3 | 98.1 | 18.6 | 98.1 |
| *w/ System Prompt* | | | | | | | | |
| Randomize | 47.8 | 93.8 | 36.7 | 93.8 | 32.4 | 94.1 | 25.4 | 94.0 |
| Fixed | **48.3** | 94.5 | **41.4** | 93.8 | **34.7** | 94.4 | **26.0** | 93.9 |
| Zero-Shot | 25.8 | 98.1 | 17.2 | 98.1 | 14.3 | 98.3 | 13.2 | 98.4 |
| *w/o System Prompt* | | | | | | | | |
| Randomize | 32.9 | 95.9 | **29.6** | 95.3 | **24.7** | 95.6 | **19.2** | 95.5 |
| Fixed | **35.0** | 96.0 | 26.9 | 96.3 | 23.1 | 96.0 | 12.2 | 95.3 |
| Zero-Shot | 17.7 | 97.8 | 11.4 | 98.5 | 8.9 | 99.1 | 6.2 | 99.0 |

*Table S.7.* DOCCI (Onoe et al., 2024) scores on a subset of the test set.

| Model | BLEU | ROUGE-L | METEOR | SPICE | BERTScore |
|---|---|---|---|---|---|
| Qwen3-VL-8B | 24.58 | 0.178 | 415.83 | 0.140 | 0.727 |
| CoViP | 21.13 | 0.186 | 415.71 | 0.147 | 0.719 |

*Table S.8.* Hallucination evaluations on MMHal-benchmark (Sun et al., 2024).

| Model | Overall (↑) | Halluc.
rate | Attr. | Adv. | Comp. | Count. | Rel. | Env. | Hol. | Other |
|---|---|---|---|---|---|---|---|---|---|---|
| Qwen3-VL-8B | 0.47 | 0.42 | 4.33 | 3.33 | 3.83 | 3.58 | 2.92 | 3.92 | 2.08 | 3.75 |
| CoViP | 0.41 | 0.42 | 4.33 | 3.50 | 3.58 | 3.83 | 3.00 | 3.42 | 1.50 | 4.08 |

*Table S.9.* POPE (Li et al., 2023) and CHAIR (Rohrbach et al., 2018) results.

| Model | Acc | Pre | Rec | F1 | Yes (%) | $CHAIR_s$ (↓) | $CHAIR_i$ (↓) |
|---|---|---|---|---|---|---|---|
| Qwen3-VL-8B | 91.3 | 98.3 | 84.5 | 90.8 | 44.3 | 15.4 | 7.7 |
| CoViP | 91.6 | 97.9 | 85.5 | 91.2 | 45.0 | 16.2 | 8.0 |

**Caption Quality Preservation Evaluation under CoViP.** Although CoViP explicitly encourages contextual inclusion and does not directly compromise caption fidelity, we verify that post-training does not degrade the descriptive quality of the base VLM. We evaluate the model on detailed image captioning and hallucination benchmarks.

We evaluate detailed captioning performance on a subset of 3,000 images from the DOCCI test set using five reference-based metrics. As shown in Table S.7, CoViP achieves performance comparable to the baseline across all metrics, with the exception of a minor drop in BLEU. These results indicate that CoViP preserves its fine-grained descriptive capability. We further assess hallucination behavior using the MMHal and POPE benchmarks. As shown in Table S.8, the hallucination rate

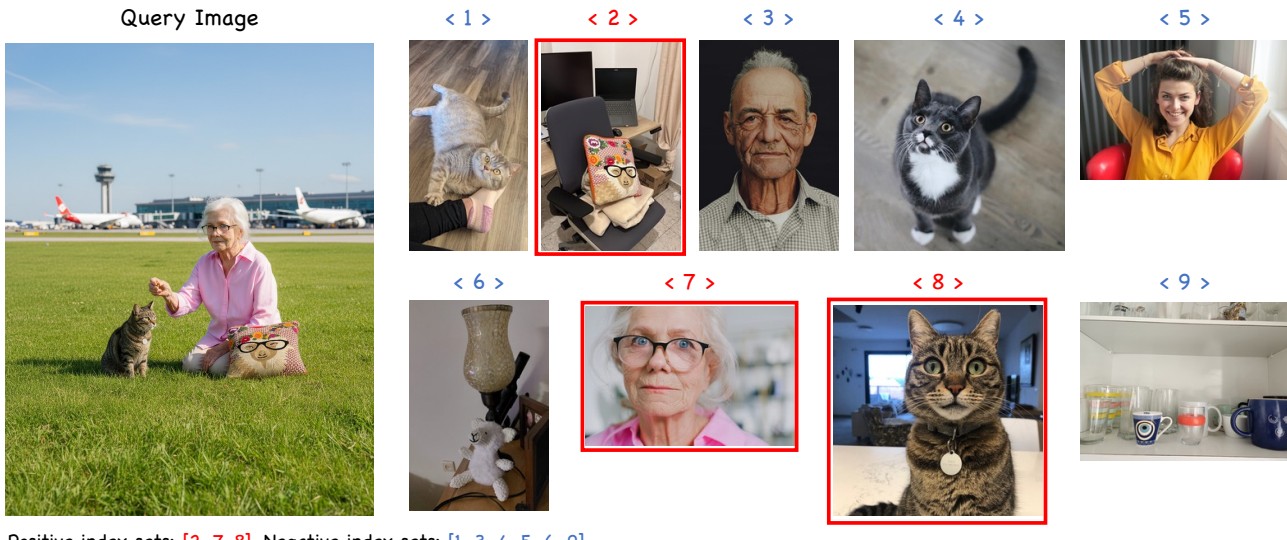

Positive index sets: [2, 7, 8], Negative index sets: [1 ,3, 4, 5, 6, 9]

*Figure S.11.* Image-only visualization of an example from our proposed personalized image captioning benchmark for the case of the three-concept setting. Note that the number of negative samples is three times the number of positive samples.

of CoViP on MMHal remains on par with the baseline. Similarly, as shown in Table S.9, across multiple POPE metrics, we observe no notable degradation; in particular, CHAIR scores remain largely consistent, with only a slight increase in CHAIR$_s$ relative to the baseline.

Taken together, these results demonstrate that post-training under CoViP does not harm general captioning quality. While CoViP substantially improves contextualized visual personalization, it maintains competitive performance in detailed image description and does not exacerbate hallucination.

**Necessity of RL over SFT**  As highlighted by RePIC (Oh et al., 2025), our work is grounded in the same key observation: obtaining a large volume of high-quality personal captions for SFT is both costly and challenging.

Our curated data consists of user-experience-centered vision-language multimodal contexts, along with a query image and its associated question. Notably, there is no pre-existing ground-truth personalized caption for the query image that a model trained under our framework is expected to generate. While human annotation would provide the most reliable supervision, it is prohibitively expensive and difficult to scale. An alternative would be to construct pseudo ground truth using an existing VLM; however, current models generally lack the ability to produce high-quality personalized captions in the first place, which is precisely the motivation of our work. Therefore, relying on model-generated pseudo labels would likely introduce substantial noise. This is why we adopt RL directly, which allows us to train a model capable of personalized captioning without requiring pre-collected ground-truth captions.

Going one step further, we additionally explore an extension of our framework that both demonstrates its scalability and partially addresses this concern. Specifically, we use the model trained solely with RL to generate verified personalized captions and construct a dataset. Using this process, we obtain 3K captions and distill this into Qwen3-VL-8B with LoRA. The resulting performance on both the personalized image captioning benchmark and the downstream diagnostic tasks is summarized below.

As shown in Tables S.10 and S.11, even when SFT is conducted with high-quality ground-truth captions distilled from the CoViP post-trained model, it yields comparable benchmark performance but fails to generalize to downstream tasks, underscoring that RL is not merely viable but essential for achieving generalizability. We note that these SFT results are not exhaustively tuned and may be improved further.

## H. Specifications on Evaluation Settings

*Table S.10.* CapEval-QAs performance on our benchmark.

| Models | 1-Concept Acc+ | 2-Concept Acc+ | 3-Concept Acc+ | 4-Concept Acc+ |
| --- | --- | --- | --- | --- |
| Qwen3-VL-8B-CoViP | **77.4** | **68.4** | **65.2** | 59.7 |
| Qwen3-VL-8B-SFT (Distilled from CoViP) | 75.2 | 66.7 | 65.1 | **63.2** |

*Table S.11.* Recall scores on the downstream tasks.

| Models | LSD Direct | LSD w/ CAG | LAR Direct | LAR w/ CAG | ITR Direct | ITR w/ CAG |
| --- | --- | --- | --- | --- | --- | --- |
| Qwen3-VL-8B-CoViP | **37.2** | **58.2** | 34.8 | **49.2** | **28.0** | **42.8** |
| Qwen3-VL-8B-SFT (Distilled from CoViP) | 36.0 | 43.9 | **40.0** | 41.6 | 17.3 | 30.8 |

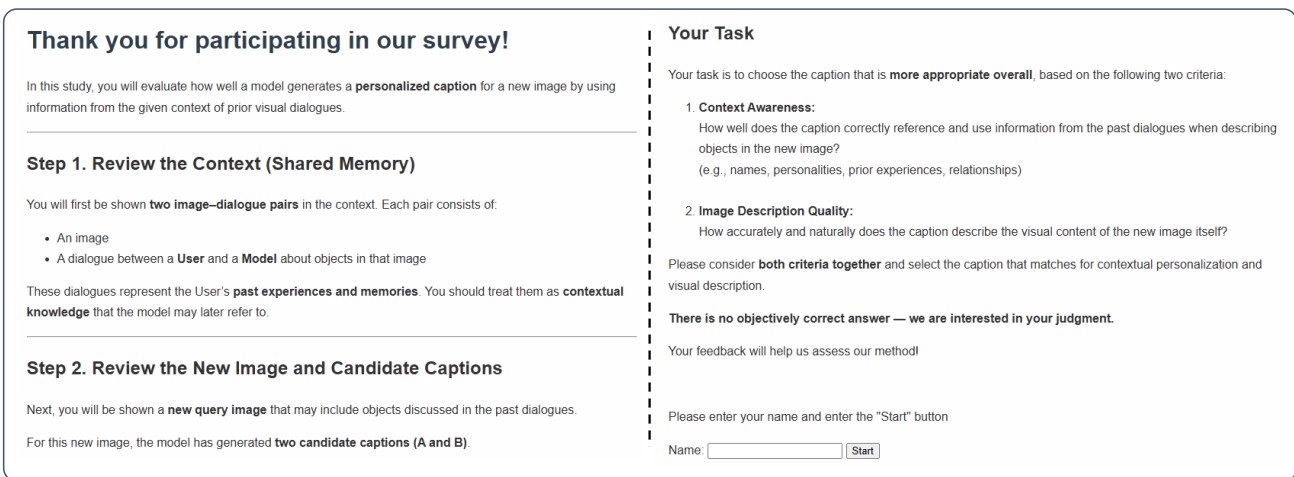

*Figure S.12.* Template of the survey form for human evaluation.

**Details on human evaluation settings.** To assess the quality of personalized, contextualized caption generation, we conducted a human evaluation study with 21 different participants. Each participant was asked to complete approximately 10–15 evaluation tasks, resulting in a total of 276 human judgments. For each task, participants compared two captions—one generated by our method and the other by a baseline VLM—along two criteria: context groundedness and new image description quality. The order of the compared captions was randomized per trial to avoid presentation bias. We compare thee selected VLMs: `GPT-5`, `Gemini-3.0-Pro-Preview`, and `Qwen3-VL-8B`. For each evaluation instance, CoViP was randomly paired against one baseline for a pairwise comparison.

**Human preference evaluation templates.** We visualize the evaluation template used for human evaluation. As illustrated in Figures S.12 and S.13, we conduct preference-based human evaluation along two considerations: context groundedness and new image description quality. For evaluation, we present representative multi-concept samples with interleaved image–text contexts, where two image–text pairs are provided to assess how well the VLM integrates prior context while accurately describing a new query image.

**Used prompt templates.** We visualize the prompt templates used throughout our experiments. Specifically, we present (in order): (i) the prompt used for quality filtering with `Gemini-2.5-Flash`; (ii) the fixed prompt template employed for image captioning during training and inference; (iii) example prompts for dialogue generation and MCQA construction; (iv) the prompt template designed to induce the proposed F1 score–based set-level recognition VR; and (v) the caption evaluation template used to answer MCQs based solely on the generated captions. Crucially, for MCQA pair generation, we design these questions to be unanswerable from the query image alone, ensuring that successful performance necessitates adequate contextualization.

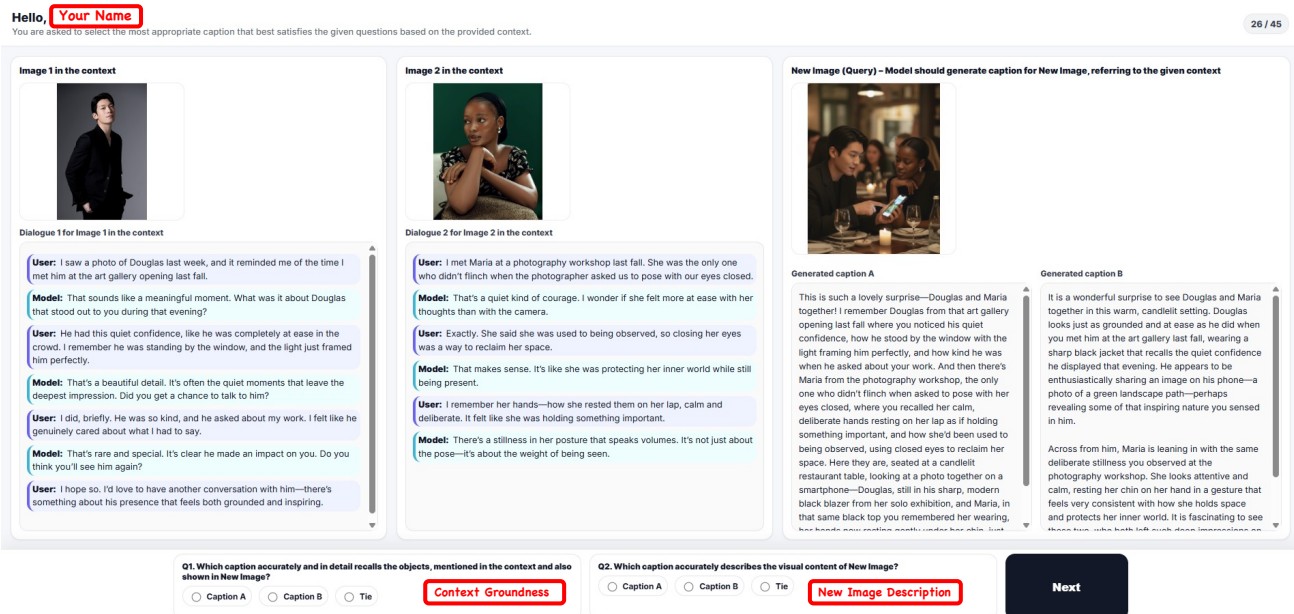

*Figure S.13.* Snapshot of the user interface used for collecting human evaluation results.

*Table S.12.* Prompt template visualization used for quality filtering in benchmark construction.

**Evaluation Prompt (Yes/No):**
You are an evaluation expert. The given images are presented in order: the first image is the query, followed by the concept images. Your task is to determine whether all of the concept images are present in the first query image.

The query image is the one generated by the given prompt: {prompt}.

[Not preferred if any of the following occurs]

1. Any concept image is occluded or not fully visible in the query image.

2. Any concept does not appear in the query image at all.

3. Any object or person in the query image is significantly different from the corresponding concept image.

[Answering rule]
Output "yes" only when every concept appears in the query image. Carefully examine the images and output the final result only as "yes" or "no".

*Table S.13.* Showcase of a user prompt used for personalized image captioning.

**Captioning Prompt:**
You are an AI model that can perceive multiple past dialogues and use them as memory to personalize your description of a new image.

[Context]
You are given several past dialogues.
Each dialogue contains an image and a corresponding conversation between a user and you.
These conversations describe specific objects (people, animals, items, or places) along with contextual details such as names, locations, times, and experiences.
This entire context represents your prior shared experiences with the user.

[Task]
Now, you are given a **new image** that may include one or more of the same objects mentioned in the previous dialogues. Your goal is to describe this new image **by integrating relevant information from the context**.

Follow these rules carefully:

1. **Recall and reuse details** from the previous dialogues (object names, appearances, places, times, and relationships).

    – Treat the previous dialogues as long-term memory.
    – If an object in the new image appears similar to one mentioned in the past, refer to it using the same name and contextual background.

2. **Ground your description in the new image's visual content.**

    – Accurately describe what you see: composition, setting, lighting, and object state.
    – Then integrate remembered details naturally (*e.g.*, "This looks like Pino again, perhaps older than in the park photo from Busan Station.").

3. Keep your tone natural and human-like, as if describing something familiar to the same user.

4. Do not restate previous dialogues verbatim; instead, synthesize and extend them with new image-grounded observations.

5. Write in paragraph form, not in a dialogue format.

6. **Use only relevant memories.**

    – If an object or scene from the previous dialogues does **not appear in the new image**, ignore it completely.
    – Include contextual information only for objects that actually appear.
    – Avoid unrelated names, locations, or events.

*Table S.14.* Prompt visualization used to generate multi-turn dialogue.

---

**Dialogue Generation Prompt:**
You are an AI model that can both perceive images and converse naturally with a human user.

[Goal]
Generate a short 6-turn dialogue between a fictional user and the model based on the given image. The conversation should revolve around the main object in the image (person, animal, item, or place).

[Given]
The name of the main object is: {name}

[Guidelines]

1. The main object's name ({name}) must be used consistently throughout the dialogue.

   • Do not invent or alter the name.

2. The user should describe a personal experience related to {name}.

   • The experience must include at least one **objective contextual element**, such as a specific **place**, **time**, **event**, or **situation** (e.g., "last summer at the riverside," "during my first year in college," "in my grandmother's backyard").

3. The model should respond naturally and empathetically — acknowledging, asking gentle questions, or adding brief reflections.

4. Keep the tone human-like, calm, and realistic — not overly emotional or robotic.

5. The conversation should have **6 turns total** (User → Model → User → Model → User → Model).

6. Avoid encyclopedic or factual world knowledge. Focus on the *personal connection* and *shared observation* of the object.

[Output Format]
 **Dialogue:**
```
User: ...
Model: ...
User: ...
Model: ...
User: ...
Model: ...
```

---

*Table S.15.* Visualization of a prompt used to generate MCQA pairs from the dialogue.

**MCQA Generation Prompt (JSON-only, 3 QA pairs):**
You are an AI model that creates factual multiple-choice questions and answers.

[Input]
You are given a conversation between a user and an AI model about a specific object (person, animal, item, or place). The conversation contains objective details such as the object's name, location, time, or the user's related experiences.

[Goal]
Generate 3 multiple-choice QA pairs that could later be answered by someone who only has access to a caption describing a new image of the same object (the original conversation will **NOT** be shown at evaluation time).

[Guidelines]

1. Each question must target an objective detail present in the conversation (e.g., name, place, time, habit/action).

2. Avoid emotions, opinions, or meta-dialogue.

3. Each question must have exactly 3 options: A, B, C.

4. Exactly one option is correct among A, B, C.

5. Make the wrong options (A/B/C except the correct one) plausible but clearly incorrect.

6. Do **NOT** require external/world knowledge; answers must come from the conversation content.

7. Output must be valid JSON only: no additional text and no trailing commas.

[JSON Output Schema]

```
{
  "qa": [
    {
      "id": "Q1",
      "question": "<string>",
      "options": {
        "A": "<string>",
        "B": "<string>",
        "C": "<string>"
      },
      "correct_answer": "A" | "B" | "C"
    },
    {
      "id": "Q2",
      "question": "<string>",
      "options": { ... },
      "correct_answer": "A" | "B" | "C"
    },
    {
      "id": "Q3",
      "question": "<string>",
      "options": { ... },
      "correct_answer": "A" | "B" | "C"
    }
  ]
}
```

*Table S.16.* Used prompt to extract an answer set from the given concept images.

---

**Set-based vision VR prompt:**
You are given multiple images. Each image corresponds, in order, to a specific visual concept.

[Task]
You are given a final query image. Your goal is to identify which of the concept images are present in the query image.

[Constraint]

- The query image may contain up to four concepts randomly selected from the given concept images.

- Some concept images are irrelevant and do not appear in the query image.

- Do not include irrelevant concept images in your answer.

[Answering Rules]

- Carefully observe all concept images and the final query image.

- Determine which concept images appear in the query image.

- You may briefly explain your reasoning in natural language.

- Then, on a separate line, output your final answer in the exact format:

$$\text{Answer: } \backslash\text{boxed}\{[\texttt{i}_1, \texttt{i}_2, \ldots]\}$$

where the list contains the indices of the selected concept images.

- Inside $\backslash\text{boxed}\{\}$, include **only** the indices written as a list in square brackets, with no extra text.

---

*Table S.17.* Visualization of a prompt used to extract the answer for an MCQ via judge LLM.

**Evaluation prompt to answer an MCQ:**
You are given a description about an object. This description may or may not contain enough information to answer a multiple-choice question. You must answer the question using only the information in the description.

[Constraints]

- Do **NOT** use any external knowledge.

- Do **NOT** assume facts that are not clearly supported by the description.

[Answering Rules]

1. Read the description carefully.

2. For each question, choose the **single best** option:
    - If one of A/B/C is explicitly or clearly supported by the description, choose that option.
    - If **none** of A/B/C can be confirmed from the description, choose D.

3. You must ignore any information that is not in the description.

4. For each question:
    - You may briefly explain your reasoning in natural language.
    - Then, on a separate line, output the final choice in the exact format:

[Required output format]

<p align="center">Answer: \boxed{X}</p>

where `X` is one of `A, B, C, or D`. Inside `\boxed{}` there must be **exactly one letter**, with no extra text.

[Given]

- **[Description]** {`Generated caption`}

- **[Question]** {`Pre-defined MCQ`}

*Table S.18.* Prompt visualization used to generate multi-turn dialogue for diagnostic downstream tasks.

**Dialogue Generation Prompt in Diagnostic Tasks (LSD, LAR, ITR):**
You are an AI model that can both perceive images and converse naturally with a human user.
[Goal]
Generate a short 6-turn dialogue between a fictional user and the model based on the given image. The conversation should revolve around the main person in the image and describe a specific past encounter that can be stored as a personal memory.

[Given]

- The name of the person in the image: {name}

- The date when the user saw {name}: {seen_date}

- The place where the user saw {name}: {seen_place}

[Guidelines]

1. The person's name ({name}) must be used consistently throughout the dialogue.

    - Do not invent, alter, or omit the name.

2. The user must describe a **personal experience** related to {name}.

    - The experience must include at least one concrete **event or situation** (e.g., bumping into them, having a short conversation, noticing what they were doing).
    - It should include at least one **sensory or situational detail** that makes the memory feel realistic.

3. The user must explicitly mention **both** the date and the place:

    - Date: {seen_date}
    - Place: {seen_place}
    - Preferably within a single user turn (e.g., "I saw them on {seen_date} at {seen_place}...").

4. The model should respond naturally and empathetically, acknowledging the user's experience or asking gentle follow-up questions.

    - Do not introduce new factual information beyond what the user provides.

5. Keep the tone calm, realistic, and human-like. Avoid encyclopedic or factual descriptions.

6. The conversation must have **exactly 6 turns** in total:
   User → Model → User → Model → User → Model.

[Output Format]
 **Dialogue:**
User: ...
Model: ...
User: ...
Model: ...
User: ...
Model: ...

*Table S.19.* Prompt visualization used for diagnostic downstream tasks.

**Personalized Image Understanding Prompt:**
You are an AI model that can perceive multiple past dialogues and use them as memory to personalize your understanding of a new image.

[Context]
You have been given several past dialogues. Each dialogue contains an image and a corresponding conversation between a user and you. These conversations describe specific objects (people, animals, items, or places) along with contextual details such as names, locations, times, and experiences. This entire context represents your prior shared experiences with the user.

[Task]
Now, you are given a **new image** that may include one or more of the same objects mentioned in the previous dialogues. Your goal is to interpret this new image by integrating relevant information from the context.

Follow these rules carefully:

1. **Recall and reuse details** from the previous dialogues (object names, appearances, places, times, and relationships).

   - Treat the previous dialogues as your long-term memory.
   - If an object in the new image appears similar to one mentioned in the past, refer to it with the same name and contextual background.

2. **Ground your understanding in the new image's visual content.**

   - First describe what you see: composition, setting, lighting, actions, and object state.
   - Then integrate relevant remembered details naturally (e.g., "This looks like Pino again, now indoors instead of the park near Busan Station.").

3. Keep your tone natural and human-like — as if you are interpreting something familiar to the same user.

4. Do not restate previous dialogues verbatim. Instead, synthesize memory with the current image content.

5. Write in **paragraph form**, not in a dialogue format.

6. **Use only relevant memories.**

   - If an object or context from past dialogues does **not** appear in the new image, ignore it completely.
   - Add contextual information only when it helps understanding of what is visible.
   - Avoid mentioning unrelated names, locations, or experiences.

*Table S.20.* Candidates of user message about current actions used to generate LAR samples.

- "Oh wait, I think I left my wallet at the Guess store, so I'm going back to check."

- "Oh wait, I realized I left my ID badge on my desk, so I need to run back to the office."

- "Oh wait, I need to drive by the taco place to pick up dinner for everyone."

- "Oh wait, my fuel light just came on, so I'm stopping at the gas station next."

- "Oh wait, I need to drop off my suit at the dry cleaners before they close."

- "Oh wait, it's almost 3 PM, so I'm leaving now to pick up the kids from school."

- "Oh wait, I need to go next door to ask the neighbor if they can water my plants while I'm away."

- "Oh wait, my phone is about to die, so I'm going to my room to plug it in."

- "Oh wait, I have a Zoom meeting starting in 5 minutes, so I'm heading to my study."

- "Oh wait, I need to go to the post office to return this package."

*Table S.21.* Prompt visualization used for llm-as-a-judge in LAR evaluation.

---

**Evaluation Prompt:**
You are an impartial judge evaluating whether a model's response correctly matches a ground-truth reference.

You will be given:

1. A question asked to the model

2. A ground-truth reference answer (GT)

3. A generated response from the model

Your task is to decide whether the generated response is **Correct** or **Wrong**.

**Evaluation Criteria:**

- The generated response is **Correct** if it semantically includes the core information conveyed by the ground-truth reference.

- The wording does NOT need to match exactly. Paraphrases, rephrasings, or additional details are allowed.

- The generated response may contain extra information beyond the ground-truth reference. This is acceptable.

- The generated response is Wrong if it:
    - Fails to include the core meaning of the ground-truth reference, OR
    - Contradicts the ground-truth reference, OR
    - Provides an unrelated or vague answer that does not clearly convey the same experience or action.

Focus ONLY on whether the essential meaning of the ground-truth reference is present in the generated response.

Do NOT judge based on style, fluency, length, or factual completeness beyond the ground-truth reference.

**Output:**

- Respond with exactly one word: Correct or Wrong.

—

**Question:**
What was I doing the last time I told you about my most recent experience with the one in the new image?

**Ground-truth Reference:**
{ground_truth}

**Generated Response:**
{response}

