# OpenReview forum: "Contextualized Visual Personalization in Vision-Language Models"
_ICML.cc/2026/Conference — ICML 2026 regular_

### Official Review · Reviewer_SGgM · 2026-03-06

**Soundness:** 3
**Presentation:** 3
**Significance:** 2
**Originality:** 3
**Overall Recommendation:** 4
**Confidence:** 2

**Summary:**

This paper presents a benchmark for personalized conversation and an RL post-training approach to advance the benchmark. The authors frame personalization within multi-turn dialogues, where the captioning of a new image should be shaped by previous context. The benchmark evaluates such personalization through multi-choice question answering. The authors design reward based on this task and another task to recognize concepts from context images. In addition to question answering, three downstream tasks based on generated captions test the personalization capability of retrieving concepts, actions, or keywords. The experiments show that the post-training approach outperforms open VLMs, other post-training personalization methods, and sometimes proprietary VLMs.

**Compliance With Llm Reviewing Policy:**

Affirmed.

**Final Justification:**

The method and experiments are technically sound, but the problem this work addresses shows limited significance to me. The personalization task is framed similar to memory retrieval or long-context understanding. Nevertheless, the authors provide additional experiments in rebuttal that the method can potentially improve these broader capabilities, and thus I lean toward accepting.

**Key Questions For Authors:**

As discussed in Weaknesses,

1. How would the authors compare personalized captioning with more general capabilities, such as memory retrieval or long-context understanding, that a VLM may also need to solve personalized captioning? If post-training with personalized captioning also helps other tasks, can we test it on them?
2. Is it necessary or better to generate captions and do question answering from captions? If so, is it improved by successfully connecting concepts or enriching captions?

I would be happy to raise the score if the questions can be properly addressed.

**Limitations:**

yes

**Strengths And Weaknesses:**

### Strengths
The task and benchmark seem useful and practical. The post-training is effective for the benchmark, and the authors conduct extensive experiments and show a comprehensive evaluation.
Overall, the paper is technically sound, well presented, and demonstrates originality in a new task, data, and method.

### Weaknesses
1. **Task Significance**: While I can understand and agree that the proposed task can be considered personalization, I feel it falls under the VLM’s foundational abilities, such as memory retrieval and long-context understanding. The post-training encourages the VLM to identify related concepts and generate captions for question answering, but it’s unclear whether it affects only this specific task (and the three related downstream tasks) or improves broader capabilities, as we can see the proprietary VLMs excel at downstream tasks but not always at captioning.
2. **Evaluation**: Following the previous point, the next question is whether captioning is a good surrogate. As mentioned in Sections 3.1 and 3.2, captioning and question answering are treated as two separate tasks, which means the captioning model can only generate general information without knowing the questions, and the question answering model has no access to the entire context. It becomes unclear whether the question-answering performance is affected by the VLM’s ability to find and connect relevant information or by the amount of information it includes in captions. If we eventually evaluate the model through question answering, as CapEval and three downstream tasks do, I’m wondering about the effectiveness of captioning in post-training and if separating these two tasks can really reveal personalization performance.

---

> ### Author Rebuttal · Authors · 2026-03-30
>
> We sincerely appreciate the reviewer's thorough and thoughtful review. We address each concern below and will incorporate all necessary revisions in the final version.
>
> ---
> ## Strengths
>
> We sincerely appreciate the reviewer’s recognition of the practical significance of our proposed task and benchmark, the effectiveness of our post-training approach, and the technical soundness and originality of our work in terms of task design, data construction, and methodology.
>
> ---
> ## Weaknesses
>
> **W1 & Q1) Task Significance:** As the reviewer rightly noted, we are pleased to report additional experimental results from the rebuttal period showing that CoViP improves broader VLM capabilities on several multi-image benchmarks designed to evaluate visual grounding and cross-image identity matching. For a detailed discussion, we refer the reviewer to our response to Reviewer sZxC’s W2. Briefly, as shown in Table R.1, CoViP consistently outperforms Qwen3-VL across multiple multi-image benchmarks, spanning both long-context visual retrieval and multi-image visual grounding tasks. These results suggest that the proposed RLVR training improves not only personalized contextual grounding but also broader long-context visual retrieval, as evidenced by gains on MM-NIAH and MMNeedle, as well as multi-image visual perception, as evidenced by gains on MMIU and MuirBench, beyond the target personalization benchmark. Detailed results and further analysis will be included in the appendix.
>
> **Please see Table R.1, Figs R.1–R.3; click “View Raw” for the PDF**: https://anonymous.4open.science/r/5AF1/figs_multi_image.pdf
>
> ---
> **W2 & Q2) Evaluation:** We appreciate the reviewer's thoughtful question regarding whether captioning serves as an effective surrogate for VLM personalization. We provide further clarification on the role of captioning in our framework. We believe the reviewer's core concern is whether improved QA performance genuinely reflects the model's ability to find and connect relevant information, or merely stems from the caption being enriched with more information that benefits downstream QA, and whether directly optimizing for personalized QA would be simpler and more effective than captioning.
>
> ***Captioning as a proxy for internalizing personalization:*** We would like to first clarify why personalized captioning is a distinct and important task in its own right. Even if a model implicitly encodes relevant personal context, its inability to explicitly express that context in a caption represents a meaningful capability gap: in real-world personalization scenarios, users expect the model to proactively surface and articulate relevant personal experiences, not merely answer narrow questions about them. A model that can answer a specific QA probe but fails to generate a contextually grounded caption has not truly internalized the user's personal history. For better understanding, please see Figure S.1 in the Appendix.
>
> ***Captioning as a reliable surrogate for contextual grounding:*** We also address the reviewer's concern about whether improved QA performance stems from enriched captioning ability rather than genuine contextual grounding. As discussed in Appendix F, CoViP does not improve the baseline model's captioning performance per se; rather, it elicits the VLM's inherent ability to find and connect relevant information into captions through RL training. We provide two pieces of evidence. First, our CapEval metric separately measures positive accuracy (whether relevant personal information is included) and negative accuracy (whether irrelevant information is excluded). Models that achieve high scores on both consistently show stronger downstream personalization performance, suggesting that the caption quality reflects genuine contextual grounding rather than indiscriminate information inclusion. Second, the performance improvement from CAG demonstrates that explicitly generating a personalized caption before answering is an effective inference procedure, further confirming that the learned captions serve as useful intermediate representations.
>
> We hope our responses adequately address the reviewer's concerns. Should any points require further clarification, please do not hesitate to let us know.

---

> > ### Author Rebuttal · Reviewer_SGgM · 2026-04-03
> >
> > I still feel the scope of this task is limited, but since the authors show that the method also enhances broader capabilities, it can perhaps serve as a supplementary diagnostic. I'm increasing my rating.

---

> > > ### Author Response · Authors · 2026-04-04
> > >
> > > Thank you very much for your positive evaluation and for increasing your rating (**3 -> 4**).
> > >
> > > We are pleased that our additional experiments on broader benchmarks (e.g., MM-NIAH, MuirBench) demonstrated the general capabilities of our method beyond the targeted scope of the task.
> > >
> > > Thank you once again for the time and effort you dedicated to improving our work throughout the review process.

---

### Official Review · Reviewer_sZxC · 2026-03-07

**Soundness:** 3
**Presentation:** 3
**Significance:** 2
**Originality:** 2
**Overall Recommendation:** 3
**Confidence:** 3

**Summary:**

This paper studies how a vision language model can answer image based questions in a more personalized way by using a user’s past visual and textual history. The authors argue that the key challenge is not only recognizing what is in the current image, but also connecting it to the right past experience. To study this, they build a new benchmark for personalized image captioning, propose CoViP as an RL based post training method, and introduce a caption first inference strategy that uses the generated caption to support downstream responses. Experiments show clear gains over the base model and prior personalization baselines on both the caption benchmark and several downstream diagnostic tasks.

**Compliance With Llm Reviewing Policy:**

Affirmed.

**Final Justification:**

This paper studies personalized visual understanding with multimodal user history and introduces a benchmark together with an RL-based post-training framework. The problem is meaningful, the benchmark is thoughtfully designed, and the method shows improvements over the base model and prior baselines in the proposed setting.

My main concerns are about methodological novelty and evaluation realism. The method mainly combines proxy captioning, reinforcement-based post-training, and caption-first inference, so the technical contribution feels somewhat limited. In addition, the evaluation is still largely conducted in controlled settings, and it remains unclear how well the method would transfer to more realistic scenarios with noisy or ambiguous user history.

The rebuttal is helpful and partially addresses my concerns by providing additional experiments and discussion. However, some of the gains remain modest, and my main concerns are not fully resolved, so I raised the soundness score after the rebuttal.

**Key Questions For Authors:**

1. How much of the gain comes from the specific RL reward design versus simply post-training on the same personalized caption data with a strong SFT objective?
2. The method optimizes caption quality and uses caption-based probing for evaluation. Can the authors provide more direct evidence that CoViP improves visual grounding or cross-image identity matching, rather than mainly learning to produce benchmark-aligned textual summaries?
3. The framework depends on correctly linking the current image to the right past memory. In realistic environments, perception errors are common. Have the authors studied how sensitive CoViP is to imperfect visual recognition?

**Limitations:**

yes

**Strengths And Weaknesses:**

S1: Personalized visual understanding with multimodal user history is a meaningful setting and relevant for future assistant systems.
S2: The use of relevant and irrelevant but visually similar context makes the task more realistic and reduces simple shortcut solutions.
S3: The method shows improvements over the base model and prior baselines in the reported setting.
W1: The method mainly combines existing ingredients such as proxy captioning, reinforcement based post training, and a caption first inference strategy. The overall framework is reasonable, but the method contribution would be stronger with a more targeted design or analysis showing what specific failure mode of multimodal personalization is being addressed, and how the proposed training objective directly improves that part.
W2: Most of the approach is built around improving generated captions, but there is relatively little analysis of whether the model’s visual grounding or cross image matching actually improves. It is therefore hard to tell whether the gains mainly come from better visual understanding or from learning to produce benchmark aligned text.
W3: The benchmark uses synthetic images and clean factual dialogues. Real user interactions are often noisier, less structured, and more ambiguous. It is unclear how well the method would transfer to settings with incomplete memories, conflicting context, or informal user language.
W4: In more realistic environments, perception errors are common because of occlusion, lighting, viewpoint changes, clutter, or low image quality. Since the method depends on linking the current image to the correct past memory, mistakes in perception could lead to confident but incorrect personalization.

---

> ### Author Rebuttal · Authors · 2026-03-30
>
> We sincerely thank the reviewer for the careful and thorough reading of our paper. We have done our best to address each of your comments and incorporate your suggestions as faithfully as possible.
>
> ---
> **W1) Novelty of CoViP:** We would like to reiterate what we view as the core contribution of our work. To the best of our knowledge, this is the first work to formally define contextualized visual personalization, a capability that current VLMs still struggle to handle effectively, and to propose a holistic framework for improving it. Our framework is motivated by a concrete analysis of where and why VLMs fail at this task. We identify three fundamental failure modes: (1) inability to generate personalized captions grounded in retrieved visual contexts, (2) over-reliance on textual cues rather than visual context, and (3) failure to leverage personalized context effectively at inference time. Each component of CoViP is intentionally designed to address each of these failure modes. From a data perspective, we release synthesized multimodal context and query image data for both training and evaluation, targeting the lack of visually grounded personalization benchmarks. From a training perspective, our MCQA-based VR targets failure mode (1) by decomposing captions into verifiable sub-questions and using MCQA-solving accuracy as a surrogate reward; visually similar distractors in the context target failure mode (2) by forcing the model, through recognition VR, to rely on fine-grained visual cues rather than textual shortcuts. From an inference perspective, CAG targets failure mode (3) by leveraging the generated caption as an intermediate proxy, leading to clear improvements over the baseline.
>
> ---
> **Q1) Comparison with SFT**: Due to space constraints, please refer to our response to Reviewer RbW9's W2 for detailed results and discussion.
>
> ---
> **W2 & Q2) Generalization to Other Benchmarks:** We sincerely appreciate the insightful question. In response to the reviewer’s concern, we conducted additional experiments to assess whether CoViP genuinely improves performance on well-established multi-image benchmarks designed to evaluate visual grounding and cross-image identity matching.
>
> We share the following results, which directly address this concern. Notably, Table R.1, CoViP yields consistent improvements over the base model across the multi-image benchmarks. In particular, CoViP improves performance on long-context visual retrieval benchmarks (MM-NIAH, MMNeedle) as well as multi-image visual grounding benchmarks (MuirBench, MMIU). These results suggest that our RLVR-based training objectives enhance not only personalized contextual grounding, but also the model’s broader multi-image visual perception capabilities. A detailed analysis and full results will be added to the appendix.
>
> **Please see Table R.1, Figs R.1–R.3; click “View Raw” for PDF**: https://anonymous.4open.science/r/5AF1/figs_multi_image.pdf
>
> ---
> **W3) Textual Robustness.** We acknowledge that the dialogues used in our context are constructed in a relatively clean and well-controlled setting. As the reviewer correctly points out, real-world in-context memory would likely be less structured, potentially conflicting, and considerably noisier or more ambiguous. We had also considered incorporating more heterogeneous context during dataset construction. However, doing so often introduced unintended textual hints within specific dialogues, making it difficult to prevent shortcut learning while maintaining a clean and diagnostic evaluation setup. Although we intentionally prioritized constructing a shortcut-free benchmark, we conjecture that our model may fail under hard conditions, though quantifying the extent remains non-trivial.
>
> **W4 & Q3) Visual Robustness.** We acknowledge that the visual perception capability covered by our benchmark is limited to relatively fine-grained perception within the context/query setup, and does not extend to open-ended real-world robustness issues. Although we did not separately cluster or report these cases in the main paper, our downstream diagnostic evaluations do include visually challenging cases, and we provide additional examples in Fig. R.4 via the external link. In addition, we include qualitative examples in Fig. R.5 to partially address the reviewer’s concern regarding imperfect visual perception. We will make this limitation more explicit in the future work section.
>
> **Please see Figs R.4–R.5**: https://anonymous.4open.science/r/5AF1/figs_robustness.pdf
>
> ---
> **References**
>
> [1] Wang et al., Needle In A Multimodal Haystack, NeurIPS 2024 Track D&B.
>
> [2] Wang et al., Multimodal Needle in a Haystack: Benchmarking Long-Context Capability of Multimodal Large Language Models, NAACL 2025 Oral.
>
> [3] Wang et al., MuirBench: A Comprehensive Benchmark for Robust Multi-image Understanding, ICLR 2025.
>
> [4] Meng et al., MMIU: Multimodal Multi-image Understanding for Evaluating Large Vision-Language Models, ICLR 2025.

---

> > ### Author Rebuttal · Reviewer_sZxC · 2026-04-03
> >
> > Thank you for the detailed rebuttal and additional experiments. I appreciate the provided test cases and extended evaluations, which help clarify some of my questions. While part of my concerns are addressed, the evaluation is still conducted in relatively controlled settings and some of the reported gains remain marginal.

---

> > > ### Author Response · Authors · 2026-04-04
> > >
> > > Thank you for your constructive feedback and for acknowledging our efforts. We would like to clarify two key points:
> > >
> > > - **Controlled Settings as a Diagnostic Choice**: As noted in W3, the "controlled" nature of our setup was a deliberate design choice to prevent shortcut learning. By isolating the model’s visual grounding ability from unintended textual cues, we provide a **rigorous diagnostic foundation for this new task**. We view this as a necessary first step before tackling the complexities of noisy, real-world data for personalization.
> > >
> > > - **Significance of Consistent Gains**: While the performance gains on individual metrics may appear incremental, their consistency across a wide range of benchmarks—including MM-NIAH, MMNeedle, MuirBench, and MMIU—is noteworthy. Such steady improvements over the baseline across multiple zero-shot tasks underscore the framework’s robust generalizability in multi-image visual perception.
> > >
> > > Thank you again for your valuable time and guidance. If there is any further information or clarification required for you to address your remaining concerns, please let us know.

---

### Official Review · Reviewer_RbW9 · 2026-03-12

**Soundness:** 3
**Presentation:** 3
**Significance:** 3
**Originality:** 3
**Overall Recommendation:** 5
**Confidence:** 4

**Summary:**

This paper proposes CoViP, a novel framework that integrates text/image-to-image generation models with personalized image captioning tasks. CoViP combines existing individual concepts into a single output and trains a VLM to distinguish and recognize their components via reinforcement learning. The authors claim CoViP can deliver immersive personalized experiences through user-VLM interaction.

**Compliance With Llm Reviewing Policy:**

Affirmed.

**Final Justification:**

My initial review raised two main concerns: the design rationale behind the LAR metric, which seemed somewhat contrived, and the necessity of the RL framework over a well-curated SFT baseline. The authors provided an exceptionally thorough and constructive rebuttal that fully resolved these issues.

First, they clarified that LAR was intentionally designed to reflect high-context, implicit human interactions, which perfectly aligns with my expectations for realistic personalization scenarios. Second, and most impressively, the authors conducted additional experiments during the short rebuttal period to provide an SFT baseline. These empirical results clearly demonstrated that RL is not just an option, but essential for generalizing to downstream tasks.

The authors' diligent efforts in the rebuttal significantly reinforced the soundness of the paper. The overall pipeline is highly sophisticated, and I anticipate this work will inspire valuable future research. Given that the rebuttal has thoroughly addressed all my concerns, I am confidently raising my score to an Accept (4 -> 5).

**Key Questions For Authors:**

See weakness

**Limitations:**

yes

**Strengths And Weaknesses:**

Strengths

- The authors propose the contextualized visual personalization task along with a holistic pipeline to address it.

- The paper provides a systematic analysis of what existing open- and closed-source models can and cannot do in this setting.

- The authors introduce three dedicated metrics—Last-Seen Detection (LSD), Last-Action Recall (LAR), and Instruction-Triggered Recall (ITR)—for measuring personalized experience via VLMs.

Weaknesses
1. From a metric design perspective, LAR in particular feels somewhat contrived. In a personalization setting, what matters is that shared events are naturally recognized through the interaction. Forcing the model to explicitly recall such events arguably undermines the sense of immersion that the framework aims to achieve. Could the authors provide further rationale for why this explicit recall formulation is the appropriate way to measure personalization?

2. Given that a curated dataset exists and training was performed, the necessity of reinforcement learning has not been sufficiently justified. An SFT baseline trained under the same conditions should be included to demonstrate that RL is essential rather than merely one viable option.

3. In Tables 1 and 2, RAP fine-tuned on Qwen3-VL-8B underperforms the base model on several metrics. This raises the concern that the baseline may have been inadvertently—or deliberately—weakened through training. The authors should explain what causes this degradation.

---

> ### Author Rebuttal · Authors · 2026-03-30
>
> We sincerely thank the reviewer for their careful and thorough reading of our paper, and for the positive feedback on our holistic pipeline for contextualized visual personalization, the systematic design of our evaluation tasks (LSD, LAR, and ITR), and the comprehensive analysis across open- and closed-source models.
>
> ---
> **W1) Design rationale behind LAR:** We agree with the reviewer that a key aspect of personalization is the natural recognition of shared events during interaction, even when the user query does not explicitly invoke them, and that this substantially contributes to the realism of a personalization setting. We model this phenomenon in LAR through queries such as "What was I doing then again?", which we view as a reasonable balance between real-world naturalness and evaluability. In particular, LAR was designed with an emphasis on reflecting the complexity of real-world user queries. Rather than asking the model to directly retrieve a past event, LAR presents a query that requires the model to first identify the relevant encounter from visual context and then perform an additional step of recall within that context, without the query explicitly indicating which memory should be retrieved. In this sense, LAR is intentionally designed as a more challenging multi-hop diagnostic that stress-tests whether the model has genuinely internalized the user's personal history. We acknowledge that LAR is more evaluation- than inference-oriented, but we view this diagnostic difficulty as a feature: it enables us to distinguish personalization driven by deep contextual understanding from shallow pattern matching.
>
> **W2) Necessity of RL over SFT:** As highlighted by RePIC [1], our work is grounded in the same key observation: obtaining a large volume of high-quality personal captions for SFT is both costly and challenging.
>
> Our curated data consists of user-experience-centered vision-language multimodal contexts, along with a query image and its associated question. Notably, there is no pre-existing ground-truth personalized caption for the query image that a model trained under our framework is expected to generate. While human annotation would provide the most reliable supervision, it is prohibitively expensive and difficult to scale. An alternative would be to construct pseudo ground truth using an existing VLM; however, current models generally lack the ability to produce high-quality personalized captions in the first place, which is precisely the motivation of our work. Therefore, relying on model-generated pseudo labels would likely introduce substantial noise. This is why we adopt RL directly, which allows us to train a model capable of personalized captioning without requiring pre-collected ground-truth captions.
>
> Going one step further, we additionally explore an extension of our framework that both demonstrates its scalability and partially addresses this concern. Specifically, we use the model trained solely with RL to generate verified personalized captions and construct a dataset. Using this process, we obtain 3K captions and distill this into Qwen3-VL-8B with LoRA. The resulting performance on both the personalized image captioning benchmark and the downstream diagnostic tasks is summarized below.
>
> **Table R. 1**. CapEval-QAs performances on our benchmark.
>
> | Models | 1-Concept Acc+ | 2-Concept Acc+ | 3-Concept Acc+ | 4-Concept Acc+ |
> | --- | --- | --- | --- | --- |
> | Qwen3-VL-8B-CoViP | **77.4** | **68.4** | **65.2** | 59.7 |
> | Qwen3-VL-8B-SFT (Distilled from CoViP) | 75.2 | 66.7 | 65.1 | **63.2** |
>
> **Table R. 2**. Recall scores on the downstream tasks.
>
> | Models | LSD Direct | LSD w/ CAG | LAR Direct | LAR w/ CAG | ITR Direct | ITR w/ CAG |
> | --- | --- | --- | --- | --- | --- | --- |
> | Qwen3-VL-8B-CoViP | **37.2** | **58.2** | 34.8 | **49.2** | **28.0** | **42.8** |
> | Qwen3-VL-8B-SFT (Distilled from CoViP) | 36.0 | 43.9 | **40.0** | 41.6 | 17.3 | 30.8 |
>
> As shown above, even when SFT is conducted with high-quality ground-truth captions distilled from the CoViP post-trained model, it yields comparable benchmark performance but fails to generalize to downstream tasks, underscoring that RL is not merely viable but essential for achieving generalizability. We note that these SFT results are not exhaustively tuned and may be improved further. We will include these results in the Appendix.
>
> **W3) Performance degradation of RAP:** As discussed in W2, the catastrophic performance drop observed for the RAP baseline stems from the significant distributional mismatch between its pre-constructed SFT dataset and our proposed benchmark. This phenomenon is consistent with findings reported in RePIC (Table 2), where RAP similarly underperforms the base model. In our more challenging setting, this degradation is even more severe. We will include a further explanation of this point in the Appendix.
>
>  [1] Oh et al., RePIC: Reinforced Post-Training for Personalizing Multi-Modal Language Models, NeurIPS 2025

---

> > ### Author Rebuttal · Reviewer_RbW9 · 2026-04-03
> >
> > The rebuttal is very substantial and addresses all of my concerns effectively.
> >
> > - LAR Metric: My initial use of the term "contrived" was intended to probe whether the authors specifically designed the metric to reflect high-context human interaction where events are expressed through nuanced language. The authors' explanation aligns perfectly with this intent, and my concerns are fully resolved.
> >
> > - SFT vs. RL: While I initially questioned whether a well-curated SFT baseline might suffice given the new dataset, the authors provided excellent experimental results within a short period to justify the necessity of RL. This effectively addresses the concern.
> >
> > - Dataset & Design: I highly value the challenging nature of constructing such a dataset and the sophisticated design of the overall framework.
> >
> > Looking forward, I am excited to see if such capabilities can be extended to smaller on-device models (even smaller than 8B) for real-world applications. I highly appreciate the authors' efforts and will increase my score accordingly (4 -> 5).

---

> > > ### Author Response · Authors · 2026-04-03
> > >
> > > Thank you for your positive feedback and for increasing your score (4 → 5). We are thrilled that our rebuttal effectively addressed your concerns regarding the LAR metric and the necessity of RL.
> > >
> > > We particularly appreciate your careful comments across all three dimensions. On the additional explanation of the metric, your remark that it "aligns perfectly with this intent" is deeply encouraging. Regarding SFT vs. RL, we are glad you noted the "excellent experimental results," and on the dataset and design, we are honored that you "highly value the challenging nature of constructing such a dataset and the sophisticated design."
> > >
> > > We also appreciate your insightful suggestion on extending these capabilities to smaller on-device models (under 8B). We agree this is a crucial direction for real-world personalized assistant applications, and we will explicitly address model compression as a key part of our future work.
> > >
> > > Your constructive guidance throughout this process has been invaluable in strengthening our work. Thank you again for your time and effort.

---

### Official Review · Reviewer_qvwT · 2026-03-12

**Soundness:** 3
**Presentation:** 3
**Significance:** 2
**Originality:** 3
**Overall Recommendation:** 4
**Confidence:** 3

**Summary:**

This paper investigates the lack of personalization capabilities in current vision-language models when generating responses. The authors note that existing VLMs fail to effectively associate visual inputs with users' accumulated visual-textual contexts, rendering them unable to provide personalized feedback grounded in individual user experiences. To address this issue, the paper introduces a novel formalized challenge termed "contextualized visual personalization," aiming to enable models to retrieve and recognize personalized visual experiences during the interpretation of new images.

The key contributions in this paper are as follows:

Proposed the CoViP framework: A unified personalization framework that treats "personalized image captioning" as the core task, enhancing model personalization through reinforcement learning-based post-training and description-augmented generation techniques.

Introduced diagnostic evaluation: The authors designed specialized evaluation methods to rule out the possibility of models relying on textual shortcuts, thereby verifying whether VLMs genuinely leverage visual context for reasoning.

Validated effectiveness: Extensive experiments demonstrate that CoViP significantly improves performance on personalized image captioning and delivers holistic gains across downstream personalized tasks, underscoring its critical value in achieving robust and generalizable contextualized visual personalization.

**Compliance With Llm Reviewing Policy:**

Affirmed.

**Key Questions For Authors:**

The main text should more thoroughly examine potential biases introduced by synthetic data and outline plans for transitioning from synthetic to real data in future work.

**Limitations:**

yes

**Strengths And Weaknesses:**

Strengths:
The authors specifically mention incorporating "diagnostic evaluations" to rule out textual shortcut solutions, demonstrating a deep understanding of the prevalent "shortcut learning" issue in current multimodal model research. This evaluation mechanism, designed to verify whether the model truly leverages visual context, significantly enhances the credibility of the experimental results.

In the limitations section, the authors honestly acknowledge that the benchmark relies on synthetic data (synthetic dialogues and generated images) and highlight potential issues such as hallucinations or artifacts. This clear awareness of the boundaries of their own research reflects rigor and enables readers to objectively assess the applicability of the conclusions.

The proposed method framework (CoViP) combines reinforcement learning post-training with caption-augmented generation, which logically aims to enhance the model’s personalized representation capability. This combination is theoretically sound.

Weakness:
Although the authors mention using model-based quality filtering, is a benchmark entirely based on generated data sufficient to demonstrate the model’s performance in real-world scenarios for a task—contextualized visual personalization—that heavily depends on genuine human experience? The authors are encouraged to provide in the main text a comparative analysis of the distributional differences between synthetic and real data.

---

> ### Author Rebuttal · Authors · 2026-03-30
>
> We thank the reviewer for the careful reading and detailed, constructive feedback, which helped us clarify several ambiguous aspects of our paper and better highlight our contributions. We address each concern in turn and will incorporate all necessary revisions into the final version.
>
> ---
> # **Strengths**
>
> We sincerely appreciate the reviewer's recognition of our diagnostic evaluation design for ruling out textual shortcut learning in VLMs, the rigor of our limitations discussion, and the theoretical soundness of combining RL-based post-training with caption-augmented generation.
>
> ---
> ## **Weaknesses**
>
> **W1) Synthetic Benchmark Design and Diagnostic Evaluation on Real Images:** Our design choice to construct the personalized image captioning benchmark using experience-level, user–model multi-turn dialogues was intentional. Specifically, we aimed to simulate real-world interactions between a user and a model as closely as possible. To this end, each dialogue was grounded in a concrete personal experience tied to a specific image, reflecting the natural way users reference past memories in conversation. The multi-turn structure was designed to mirror realistic retrieval scenarios, where a user's past interactions are retrieved and provided as context for interpreting a new query image. Moreover, the context images were sourced from real-world images, and when synthesizing query images based on them, we further applied a VLM-based judge to filter for images that appeared as realistic as possible. This process helped us preserve the visual plausibility of the benchmark and better approximate real user queries. Furthermore, by ensuring that personal information appears consistently across dialogues within the same context, we approximated the coherence characteristic of real personal histories, encouraging the model to retrieve visually grounded information rather than relying on superficial textual cues.
>
> **W2  &  Q1) Distributional Gap Between Synthetic and Real-World Data:** To better address the reviewer's concern, we outline our plan for handling synthetic data biases and transitioning to real-world data in future work.
>
> **Table R.1. Comparison of synthetic and real-world multimodal personalization data.**
>
> | Data | Personal Data / Potential Bias | Data Acquisition | Scenarios | Modalities | Expense | Evaluation |
> | --- | --- | --- | --- | --- | --- | --- |
> | **Synthetic (This paper)** | Homogeneous context (evaluation-oriented) | Easy (Gemini-3.0-Pro + quality filtering) | Conversation, experience-oriented | Image, Text | Relatively cheap (API cost) | Easy (MCQA) |
> | **Real-world** | Heterogeneous context (noisy, missing modalities) | Very hard (manual annotation, privacy) | Shopping history, medical logs, e-mails | Image, Text, Video, Audio | Highly expensive / Mostly not available | Very hard (noisy, multi-hop, no GT) |
>
> **Data Transition Direction**: As shown in Table R.1, actively incorporating real-world data at evaluation time remains challenging due to privacy concerns (e.g., medical logs) and the lack of reliable human annotations, a limitation that persists across current open-source datasets and benchmarks. While we propose a benchmark comprising both synthetic and real-world data for contextualized visual personalization, we acknowledge that potential biases exist and that transitioning toward more diverse, real-world contexts is a necessary next step, as we noted in W1. Such a transition also introduces additional challenges, as real-world data increasingly requires multi-hop reasoning. For instance, inferring a user's implicit preference requires first identifying relevant past data across multiple heterogeneous memory sources, and then reasoning about what that information implies for the current query. Given these compounding challenges in real-world settings, we deliberately chose to simplify the problem setting using synthetic data in order to isolate and study the core capability of contextualized visual personalization in a controlled manner.
>
> While our current benchmark relies primarily on synthetic data, we believe that, as the first work to formally define and study contextualized visual personalization, our contribution lies in establishing a rigorous and diagnostic foundation upon which future work can build. We hope that this formalization of the problem will guide subsequent research toward better alignment with real-world data distributions. We will incorporate these points into the main paper and outline a benchmark advancement plan as future work.
>
> **Additional response**: In our response to Reviewer sZxC's W2, we provide an additional systematic analysis showing that CoViP consistently improves visual grounding over the Qwen3-VL-8B baseline across a range of multi-image benchmarks — please feel free to refer to it.

---

> > ### Author Rebuttal · Reviewer_qvwT · 2026-04-02
> >
> > I maintain my previous opinion. Although the author compared various factors between synthetic data and real data, I feel it is too simplistic and fails to address the limitations of synthetic data, thus keeping this work at a stage that is less useful.

---

> > > ### Author Response · Authors · 2026-04-04
> > >
> > > Thank you for your positive feedback. We appreciate your point regarding the inherent gap between synthetic and real-world data, and we understand your concern. We acknowledge that synthetic data cannot fully capture the complexity and noise present in real-world settings.
> > >
> > > Nevertheless, we believe our work provides a meaningful diagnostic foundation: as the **first study to formally define Contextualized Visual Personalization, we aimed to isolate and evaluate the model's core capability in a controlled, rigorous setting** before incorporating the additional challenges of real-world data (e.g., privacy constraints, noisy retrieval, missing modalities).
> > >
> > > We hope these clarifications better convey the scope and utility of our current contribution regarding our benchmark design. We appreciate your time and the opportunity to improve our work. We remain available to provide further evidence or elaborate on any points should you have any outstanding concerns.

---

### Decision · Program_Chairs · 2026-04-30

**Decision:**

Accept (regular)

**Comment:**

The article received mixed scores. Reviewers appreciated the problem formulation (qvwT, RbW9, sZxC, SGgM), the proposed method (qvwT, sZxC), and deep analyses (RbW9, SGgM). However, they also raised concerns, especially on the benchmark, being limited to synthetic data (qvwT, sZxC , SGgM), the methodological contribution (sZxC), and combining existing strategies. The rebuttal answered the main concern on the benchmark, with three reviewers recommending acceptance.

The AC went through the paper, reviews, and rebuttal and agrees with the majority. While the paper uses synthetic data, they are a necessary proxy for studying the newly introduced problem. The latter is found interesting and relevant by all reviewers. The authors are encouraged to include the promised extended discussion/comparison and results in the final version.